# Significant underestimation of radiative forcing by aerosol–cloud interactions derived from satellite-based methods

Hailing Jia [1,2,3], Xiaoyan Ma[1✉], Fangqun Yu [2✉] & Johannes Quaas [3]

Satellite-based estimates of radiative forcing by aerosol–cloud interactions ($RF_{aci}$) are consistently smaller than those from global models, hampering accurate projections of future climate change. Here we show that the discrepancy can be substantially reduced by correcting sampling biases induced by inherent limitations of satellite measurements, which tend to artificially discard the clouds with high cloud fraction. Those missed clouds exert a stronger cooling effect, and are more sensitive to aerosol perturbations. By accounting for the sampling biases, the magnitude of RFaci (from $-0.38$ to $-0.59\,\mathrm{W\,m^{-2}}$) increases by 55 % globally (133 % over land and 33 % over ocean). Notably, the $RF_{aci}$ further increases to $-1.09\,\mathrm{W\,m^{-2}}$ when switching total aerosol optical depth (AOD) to fine-mode AOD that is a better proxy for CCN than AOD. In contrast to previous weak satellite-based $RF_{aci}$, the improved one substantially increases (especially over land), resolving a major difference with models.

[1] Collaborative Innovation Center on Forecast and Evaluation of Meteorological Disasters, and Key Laboratory for Aerosol-Cloud-Precipitation of China Meteorological Administration, School of Atmospheric Physics, Nanjing University of Information Science & Technology, Nanjing, China. [2] Atmospheric Sciences Research Center, University at Albany, Albany, NY, USA. [3] Institute for Meteorology, Universität Leipzig, Leipzig, Germany. ✉email: xma@nuist.edu.cn; fyu@albany.edu

By acting as cloud condensation nuclei (CCN), aerosol can alter cloud properties and precipitation[1,2], thereby influencing the Earth's radiation budget and hence climate change. An increase in CCN number concentration will generate a cloud with more droplets. The consequence is scattering more solar radiation back to space, thus exerting a negative climate forcing. This is known as the cloud albedo effect or the Twomey effect[1]. Although extensive investigations have been made to quantify the radiative forcing by aerosol–cloud interaction (RF_aci), significant uncertainties remain on its magnitude[3,4].

The satellite-based RF_aci using retrievals of column aerosol and cloud properties, typically in the range of −0.2 to −0.7 W m$^{-2}$ (see refs. [5–9]), is much weaker than the modeled values of −0.3 to −1.8 W m$^{-2}$ (see ref. [3]). Though observational estimates based on metrics like aerosol mass, that are derived including additional model information, tend to generate higher RF_aci (−0.97 ± 0.23 W m$^{-2}$)[10,11], they are still generally lower than modeled values. Studies constraining numerical models with satellite observations reported smaller RF_aci values than that from models alone[12,13]. It is also noteworthy that, by putting a larger weight to satellite-based studies, the best estimate of RF_aci by IPCC decreased from −0.7 W m$^{-2}$ (see ref. [3]) to −0.45 W m$^{-2}$ (see ref. [4]). Therefore, it is essential to reconcile significant differences between satellite- and model-based RF_aci, in particular, to improve the estimates from an observational perspective.

In addition to the uncertainties of model simulations, this discrepancy may also be partly due to satellite-related issues. The most unavoidable limitation is retrieval biases in both aerosol and cloud properties, such as overestimated aerosol optical depth (AOD) due to either cloud contaminations[14] or cloud adjacency effects[15], as well as misestimated cloud effective radius (CER) thus cloud droplet number concentration ($N_d$) owing to inadequate retrievals applied to broken and/or inhomogeneous clouds[16]. Further investigations suggested that the covariation of retrieval biases in aerosol and cloud properties could incur a false correlation between them, thereby underestimating the cloud albedo effect[17]. By utilizing satellite simulators, Ma et al.[18] found that low-aerosol loading conditions are not well detected by satellites, but modeled clouds are sensitive to aerosol perturbations in these conditions, which contributes a large part to the difference of cloud susceptibilities derived from model and satellite. Also, whether the $N_d$–AOD relationship under present-day (PD) can be used to determine the preindustrial (PI) $N_d$[19,20], and whether the AOD in cloud-free regions is an adequate proxy for CCN at cloud base[21], are still under debate. If aerosol information is available at cloud base altitude, an even stronger aerosol–cloud relationship would be expected[22]. All of these tend to underestimate the RF_aci[19,23].

Our focus here is on another potential contributor to the underestimates of satellite-based RF_aci, i.e., sampling biases, which were not explored in detail previously. Passive remote sensing only allows us to retrieve aerosol properties in clear pixels. In order to collect adjacent aerosol and cloud retrievals for statistical analysis, many studies used AOD on a coarse-resolved grid (such as 1° × 1° on a latitude–longitude grid) to match cloud pixels, assuming that aerosols properties in adjacent clear areas are representative of those under cloudy conditions[5,7,24]. Nevertheless, in the case that the clouds fully cover this larger grid box, hampering any AOD retrievals, these clouds were not sampled for the analysis on a daily basis. It is thus expected that stratiform clouds with high cloud fraction ($f$) at the aggregate scale would be artificially and systematically excluded in current satellite-based investigations that link daily aerosol and cloud properties. This is problematic especially because stratiform clouds have been reported to exert much stronger aerosol indirect effects (AIE) than cumulus clouds[25,26].

Typically, the sampling biases can affect derived RF_aci values through two pathways. First, it changes the regressions between cloud quantities and AOD (i.e., $\frac{d\ln N_d}{d\ln AOD}$). All satellite-related estimates would suffer from this, including the studies that utilize pure satellite measurements[5,27,28], that involve satellite measurements but with a radiative transfer model[29], as well as that constrain models with satellite observations[12,13]. The second pathway is by altering $f$ directly, key parameters in the calculation of RF_aci (see Eq. (3) in "Methods"). This pathway is only relevant to the abovementioned pure satellite-based investigations that require the coexistence of retrievals of cloud and aerosol when computing RF_aci value.

In this study, we employ the satellite-based approach proposed by Quaas et al.[5] to reassess RF_aci by accounting for the impacts of sampling biases. The key idea is to make use of an aerosol reanalysis product that is tied to the satellite-retrieved AOD wherever it is available and also makes use of model information and thus allows for a consistent AOD estimate everywhere in space and time, including in regions that are cloud-covered. The RF_aci is also estimated by adopting fine-mode AOD (AOD_f) in addition to AOD that was commonly used in previous satellite-based investigations, as well as different anthropogenic fractions, to assess the sensitivity of the results to choices of CCN proxy and anthropogenic fraction. We find that after fixing the sampling biases the RF_aci is substantially more negative (particularly over land), along with a surprisingly similar spatial distribution to the modeled result. Also, the magnitude of RF_aci almost doubles when switching AOD to AOD_f which is a better proxy for CCN.

## Results

Most satellite-based studies on RF_aci estimates and/or aerosol–cloud correlations require the coincidence of aerosol and cloud retrievals and thus miss cloud samples in grid boxes in which no aerosol is successfully retrieved. To explore the influences of sampling biases quantitatively, analyses under different scenarios are conducted by combining cloud retrievals from the Clouds and the Earth's Radiant Energy System (CERES)[30] with the MODerate Resolution Imaging Spectroradiometer (MODIS) aerosol retrievals[31] and the Modern-Era Retrospective analysis for Research and Applications, version 2 (MERRA-2) aerosol reanalysis[32]. The coarse-resolution aerosol data (1° × 1° resolution for MODIS retrieval and 0.5° × 0.625° resolution for MERRA-2 reanalysis) are projected to the higher resolution of pixel-scale cloud observations, generating 20 × 20 km$^2$ resolution aerosol–cloud data pairs for analysis. The key idea is that the reanalysis is tied to the satellite retrievals of AOD wherever they are available (in cloud-free conditions) but also provides AOD in cloud-covered regions. Figure 1 illustrates four basic scenarios. Aero_Cld includes the samples for which both aerosol (according to the satellite sampling) and cloud retrievals are available, while Cld includes ones that only cloud retrievals are available (filling in AOD from the model information in the reanalysis). All_Cld employs the combined datasets in Aero_Cld and Cld, i.e., all available ambient clouds. Aero_Cld_Modis is the same as Aero_Cld but using MODIS AOD, which is the most common configuration in satellite-based investigations. Aero_Cld should be largely consistent with Aero_Cld_Modis as MERRA-2 assimilates MODIS AOD, but avoids retrieval artifacts at least to some extent. On the basis of Aero_Cld and All_Cld, two additional scenarios (Aero_Cld_R and Aero_Cld_C) are designed to quantify the individual contributions of changed $f$ and regression slopes ($\frac{d\ln N_d}{d\ln AOD}$) to RF_aci estimate (see the section "RF_aci estimates").

**Missed cloud samples.** Although the sampling biases have been initially noticed[6,33,34], it remains unclear what amount of data could be missed and its potential consequences, which is essential to correct the results from previous satellite-based studies. Table 1

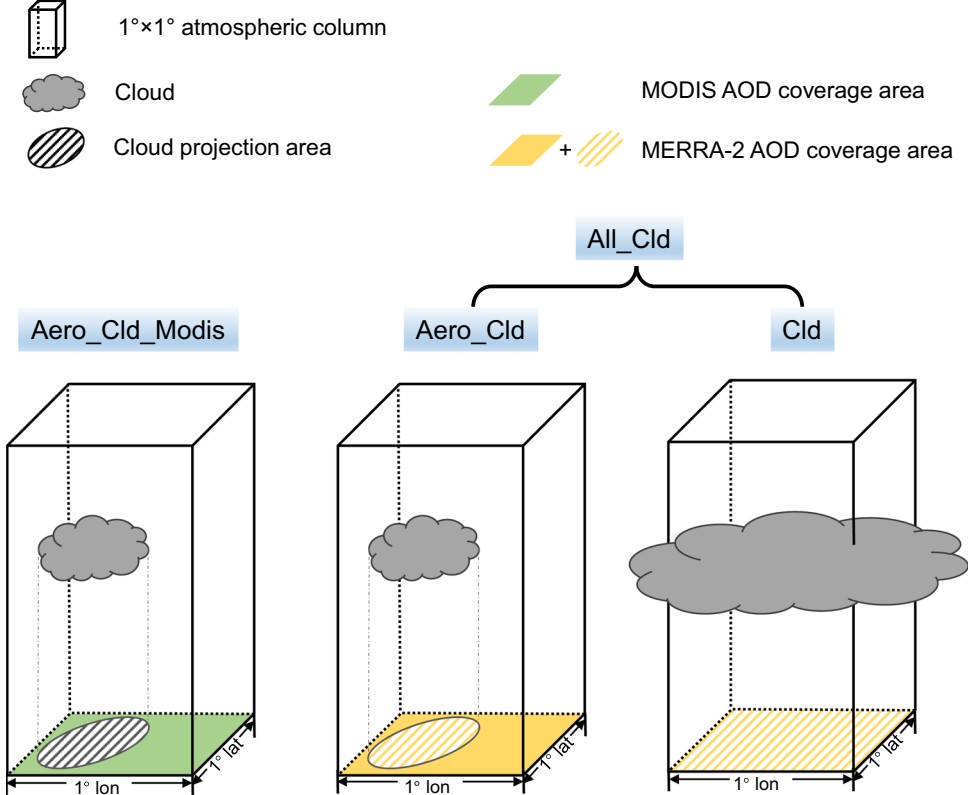

**Fig. 1 Schematic diagram of four basic scenarios in this study.** The schematic shows the combinations of clouds and its associated aerosol retrieval (green; MODIS aerosol optical depth (AOD))/reanalysis (yellow; MERRA-2 AOD) within 1° × 1° atmospheric column (cuboid) for different scenarios. Aero_Cld_Modis represents the combination of the clouds not fully covering 1° × 1° area and its adjacent MODIS aerosol retrieval. Aero_Cld includes the same cloud samples as Aero_Cld_Modis but utilizing MERRA-2 AOD. Cld scenario involves the clouds fully covering 1° × 1° area, i.e., no successful AOD retrieval so that one has to fill with re-analyzed AOD. All_Cld employs the combined datasets in Aero_Cld and Cld, including all available ambient clouds.

**Table 1 The total number of samples in All_Cld ($N_{total}$), and the number of samples ($N$), averaged cloud fraction ($f$) in Aero_Cld and Cld, respectively, for the fourteen regions over the period 2002–2018.**

| Region | $N_{total}$ (#) | $N$ (#) in Aero_Cld | $f$ (%) in Aero_Cld | $N$ (#) in Cld | $f$ (%) in Cld |
|---|---|---|---|---|---|
| NAM | 23,920,376 | 8,483,378 | 37.6 | 15,436,998 | 88.1 |
| EUR | 16,880,708 | 5,111,997 | 41.3 | 11,768,711 | 90.4 |
| ASI | 31,432,972 | 12,154,722 | 41.7 | 19,278,250 | 89.2 |
| AFR | 26,144,924 | 10,827,397 | 48.6 | 15,317,527 | 83.1 |
| SAM | 29,635,448 | 9,827,060 | 49.1 | 19,808,388 | 82 |
| OCE | 10,451,012 | 5,247,169 | 38.7 | 5,203,843 | 75.6 |
| NPO | 133,923,104 | 54,758,744 | 71.5 | 79,164,360 | 87.3 |
| NAO | 85,026,928 | 41,927,792 | 65.6 | 43,099,136 | 80.6 |
| TPO | 174,596,224 | 84,245,712 | 59.9 | 90,350,512 | 63.9 |
| TAO | 84,316,880 | 39,570,820 | 60.7 | 44,746,060 | 68.1 |
| TIO | 51,639,812 | 28,545,216 | 51.1 | 23,094,596 | 51.3 |
| SPO | 170,840,160 | 79,262,384 | 75.4 | 91,577,776 | 89.1 |
| SAO | 91,109,944 | 37,466,440 | 77.4 | 53,643,504 | 92 |
| SIO | 126,288,636 | 56,619,836 | 76.8 | 69,668,800 | 91.1 |

summarizes the number of samples ($N$; i.e., daily 20 × 20 km² resolution aerosol–cloud data pairs) and averaged $f$ for Aero_Cld and Cld for fourteen different oceanic and continental regions (see Supplementary Fig. 1 for the geographical distribution), respectively, over the period 2002–2018. It is noteworthy that $N$ for Cld is comparable to that for Aero_Cld, and even doubled in some regions over land (e.g., NAM, EUR, ASI, and SAM), indicating that more than half of the cloud samples were artificially discarded in previous satellite-based researches. Meanwhile, $f$ for Cld in each region is substantially larger than that for Aero_Cld,

implying those missed clouds also have a stronger radiative effect. After including all cloud samples, globally averaged $f$ increases relatively by 13%, compared to Aero_Cld. Spatially, significant increases of $f$ occur over land (Fig. 2), especially for the regions with strong anthropogenic emissions (e.g., NAM, ERU, ASI, and SAM). According to Eq. (3) in "Methods", such a spatial pattern would further amplify the effect of sampling biases on $RF_{aci}$.

In addition to altering average cloud properties, the sampling bias is also likely to influence the regressions between cloud

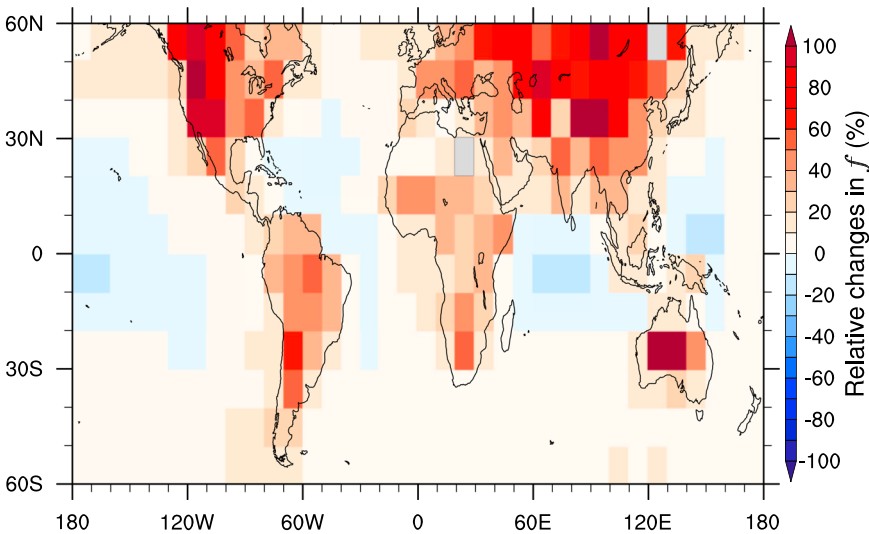

**Fig. 2 Geographical distribution of relative changes (%) in cloud fraction (f).** Changes in *f* from the scenario only including the cloud samples with successful aerosol retrievals (Aero_Cld) to the scenario including all ambient clouds (All_Cld), which are computed for each 10° × 10° grid box over the period 2002–2018.

quantities and AOD. As shown in Fig. 3a, b, the slopes of ln $N_d$ versus ln AOD are overall larger for Cld and All_Cld than that for Aero_Cld, with the exception of EUR and three regions over the Southern Ocean (SPO, SAO, and SIO). Note that the Southern Oceans only has a limited impact on the global-averaged $RF_{aci}$ due to its low anthropogenic fraction[35]. In situ observations have demonstrated that the response of $N_d$ to aerosols is more sensitive for stratiform clouds than cumuliform clouds[26]. Also, stratiform ones have smaller retrieval biases for the $N_d$ retrieval than cumulus ones[16], thus suffer less AIE underestimation caused by partly cloudy retrievals[17,25]. Thus, the larger slopes for Cld and All_Cld may be due to the inclusion of more samples with large *f*, which corresponds predominantly to stratiform clouds. Furthermore, to check if using different CCN proxies can change the above conclusion, we conduct the regression analyses based on re-analyzed aerosol index (AI) containing the information of aerosol size, and sulfate mass concentration (SO4) that eliminate the influence of aerosol swelling, which have been reported to greatly modulate the correlations between AOD and cloud quantities ($N_d$[36], *f*[37], and CER[38]) via relative humidity. Supplementary Fig. 2 shows that the results are rather similar with the AOD-based analysis, i.e., generally larger slopes for Cld and All_Cld than Aero_Cld.

For a better comparison with previous satellite-based results, it is critical to know the change of slopes induced by utilizing different AOD products (e.g., satellite versus reanalysis data). As shown in Fig. 3a, b, the slopes for Aero_Cld are ~65% higher than those for Aero_Cld_Modis over land on average, whereas no systematic difference is found over the ocean. The difference between the two tests is likely due to the retrieval biases of AOD in Aero_Cld_Modis, which have been reported to result in a serious underestimation of AIE due to the covariation of retrieval biases in AOD and CER[17]. When retrieving aerosol in cloudy pixels, AOD can be artificially overestimated due to either cloud contaminations[14] or cloud adjacency effects[15], and the over-estimation has been found to increase with *f*[14]. However, MERRA-2 AOD implements an online aerosol chemistry, radiation, and transport model and restricts the assimilation of MODIS to the pixels of *f* <70% to ensure a less biased source[32], so that it can provide more physically reasonable aerosol distributions to avoid the spuriously high AOD near clouds, thereby largely reducing the retrieval artifacts as seen in MODIS AOD. As

expected, MODIS AOD agrees well with MERRA-2 for the clear sky, but the former is indeed systematically higher than the later in the presence of clouds (Supplementary Fig. 3), along with an increasing difference as *f* increases (Supplementary Fig. 4). Also, the response of the difference to *f* over land is overall more sensitive than that over the ocean, implying that more serious retrieval artifacts occur over land, which might explain the larger discrepancy of slopes between Aero_Cld and Aero_Cld_Modis over land (Fig. 3a, b). Given all that, the re-analyzed AOD may be a better option for correlating with cloud quantities.

For Aero_Cld_Modis, the slopes of ln $N_d$ versus ln AOD over land (0.08–0.33) are significantly lower than those over the ocean (0.14–0.51), which is consistent with previous satellite observations analyses[5,7]. The other three tests using MERRA-2 AOD, however, show weaker land–sea contrast. Also, the modeled land–sea contrast was reported to be much weaker than the satellite-observed one[39,40]. It is thus likely that the land–sea contrast of slopes might have been overestimated by previous studies relying purely on satellite observations.

AI and $AOD_f$ are believed as better proxies for the CCN than total AOD, as they are representative of fine-mode aerosol particles, which contribute the most to CCN number concentration[19]. However, the analysis based on MODIS AI or $AOD_f$ is not conducted in this study due to the poor skill in retrieving aerosol size parameters over land[31]. As an alternative, the sparser but more reliable POLarization and Directionality of Earth's Reflectance-3 (POLDER-3) $AOD_f$[41] is used here, which has been extensively validated with ground-based observations over land[42,43]. Since the sparseness of POLDER-3 $AOD_f$ would result in a serious lack of data, we also employ MERRA-2 $AOD_f$ with full spatial and temporal coverage, which is calculated as the sum of AODs of sulfate, black carbon, organic aerosols, and 30% sea salt aerosols[6]. Figure 3c, d compares the slopes of ln $N_d$ -ln AOD and ln $N_d$ -ln $AOD_f$ based on both POLDER-3 retrievals and MERRA-2 reanalysis, respectively. The results show that there is no systematic difference between AOD- and $AOD_f$-based slopes over lands. An exception is the dust source region (AFR), where the slope of ln $N_d$ -ln $AOD_f$ is higher presumably due to eliminating the influence of dust events. Over most ocean regions, in turn, the slopes with respect to $AOD_f$ are systematically larger than the ones with respect to AOD with only a few exceptions. This can be found in both MERRA-2- and POLDER-3-based

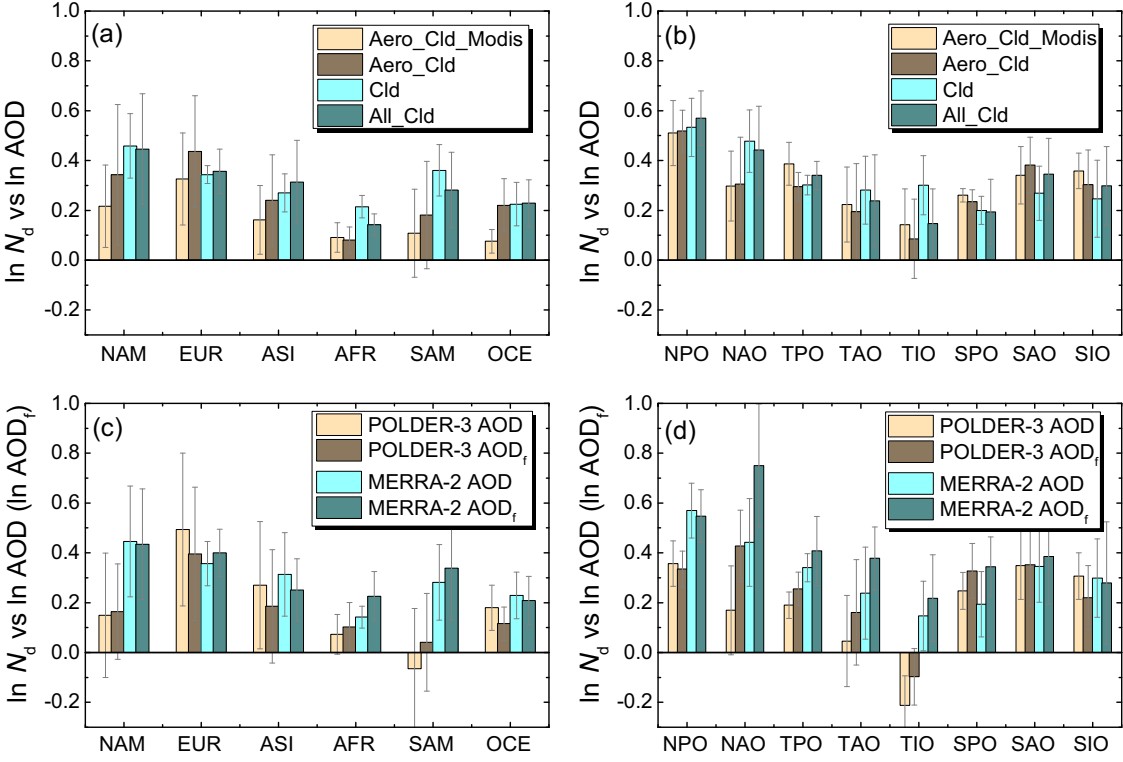

**Fig. 3 Annual averaged slopes of the linear regressions between the logarithm of cloud droplet number concentration ($N_d$) and those of aerosol optical depth (AOD) and fine-mode AOD, i.e., ln $N_d$ versus ln AOD (AOD$_f$), for different scenarios. a, b** The slopes calculated from retrieved AOD-$N_d$ data pairs (Aero_Cld_Modis), and from MERRA-2 re-analyzed AOD and CERES-retrieved $N_d$ data pairs for the scenario that aerosol and cloud retrievals are simultaneously successful (Aero_Cld), the scenario that only cloud retrievals are successful (Cld), and the scenario including all ambient clouds (All_Cld). The slopes calculated by POLDER-3 AOD, POLDER-3 fine-mode AOD (AOD$_f$), MERRA-2 AOD (same as All_Cld), and MERRA-2 AOD$_f$ are also shown (**c, d**). The annual averaged slope here is the average of the monthly slopes. The standard deviation of the inter-monthly variability of the regression slopes is shown as an error bar. A total of $12 \times 14 \times 7 = 1176$ linear regressions (for twelve months, fourteen regions and seven scenarios) were conducted, among which the slopes for 95% cases are at a statistically significant level (according to a Student's $t$ test, $\alpha = 0.01$).

results. It is noted that POLDER-3-based slopes are generally smaller than MERRA-2-based ones, possibly due to the serious sampling bias in the former. Therefore, to account for both sampling biases and contribution of fine-mode aerosols, RF$_{aci}$ based on MERRA-2 AOD$_f$ (i.e., same as All_Cld but using AOD$_f$) will be estimated in the next section.

**RF$_{aci}$ estimates**. To compute RF$_{aci}$, a change of AOD (AOD$_f$) from PI to PD is necessary, which is obtained from GEOS-Chem-APM[44] simulations basically (see "Methods"). Figure 4a–d shows the spatial distributions of RF$_{aci}$ for Aero_Cld_Modis, Aero_Cld, Cld, and All_Cld, respectively. The corresponding values for different regions are summarized in Supplementary Table 1. For ease of comparison, the modeled one from Yu et al.[45] is shown here as a reference (Fig. 4e). For Aero_Cld_Modis (Fig. 4a), which derives RF$_{aci}$ in the same manner as previous satellite-based studies[5,7] but with the updated dataset, the global annual average RF$_{aci}$ is estimated as −0.36 W m$^{-2}$ (−0.24 W m$^{-2}$ over land and −0.4 W m$^{-2}$ over ocean). The slightly higher value than earlier estimates, −0.2 W m$^{-2}$ by Quaas et al.[5] and −0.34 W m$^{-2}$ by Ma et al.[7], is likely caused by the improvement of retrieval algorithms in the updated dataset. Despite a slight increase, RF$_{aci}$ is still considerably lower than that from models (the best estimate of −0.7 W m$^{-2}$ (see ref. [3])), and shows a different spatial pattern compared to models (Fig. 4e, and also see Boucher and Pham[46]; Chen et al.[47]; Déandreis et al.[48]), i.e., much smaller RF$_{aci}$ over land with strong anthropogenic emissions (e.g., NAM, EUR, and ASI; Fig. 4a). By instead using MERRA-2 AOD (Aero_Cld),

which enhances the slopes of ln $N_d$ versus ln AOD over land compared to Aero_Cld_Modis (Fig. 3), the estimated global RF$_{aci}$ correspondingly increases to −0.38 W m$^{-2}$ (−0.33 W m$^{-2}$ over land and −0.4 W m$^{-2}$ over ocean).

As discussed in the last section, the sampling biases can induce remarkable impacts on both cloud properties and susceptibility of $N_d$ to AOD, which are the key terms to calculate RF$_{aci}$ (Eq. (3) in "Methods"). Figure 4c illustrates that the clouds missed by the satellite-based method have a much stronger RF$_{aci}$ than those that actually have been analyzed (−0.75 W m$^{-2}$ versus −0.38 W m$^{-2}$), particularly over land (−1.22 W m$^{-2}$ versus −0.33 W m$^{-2}$), which is attributable to the joint impact of the increased $f$ (Fig. 2) and slopes of ln $N_d$ versus ln AOD (Fig. 3a, b). After including all cloud samples into the calculation, which is consistent with what models do, the estimated RF$_{aci}$ increases to −0.59 W m$^{-2}$ (−0.77 W m$^{-2}$ over land and −0.53 W m$^{-2}$ over ocean), along with a surprisingly similar spatial distribution to the modeled result (Fig. 4d, e), i.e., the maximum values over major continents and followed by immediate outflow regions. There also exists a significant contrast in RF$_{aci}$ between hemispheres, i.e., much stronger in the Northern (−0.98 W m$^{-2}$) than in the Southern hemisphere (−0.19 W m$^{-2}$), where anthropogenic emissions are weaker.

By accounting for the effect of sampling biases, the estimated RF$_{aci}$ has increased by 55% (from Aero_Cld to All_Cld) on a global average (133% over land and 33% over ocean). The most evident enhancements are over the land areas with strong anthropogenic aerosol emissions, e.g., NAM (−0.91 W m$^{-2}$), EUR (−0.41 W m$^{-2}$), and ASI (−0.72 W m$^{-2}$). As mentioned in the last section, both

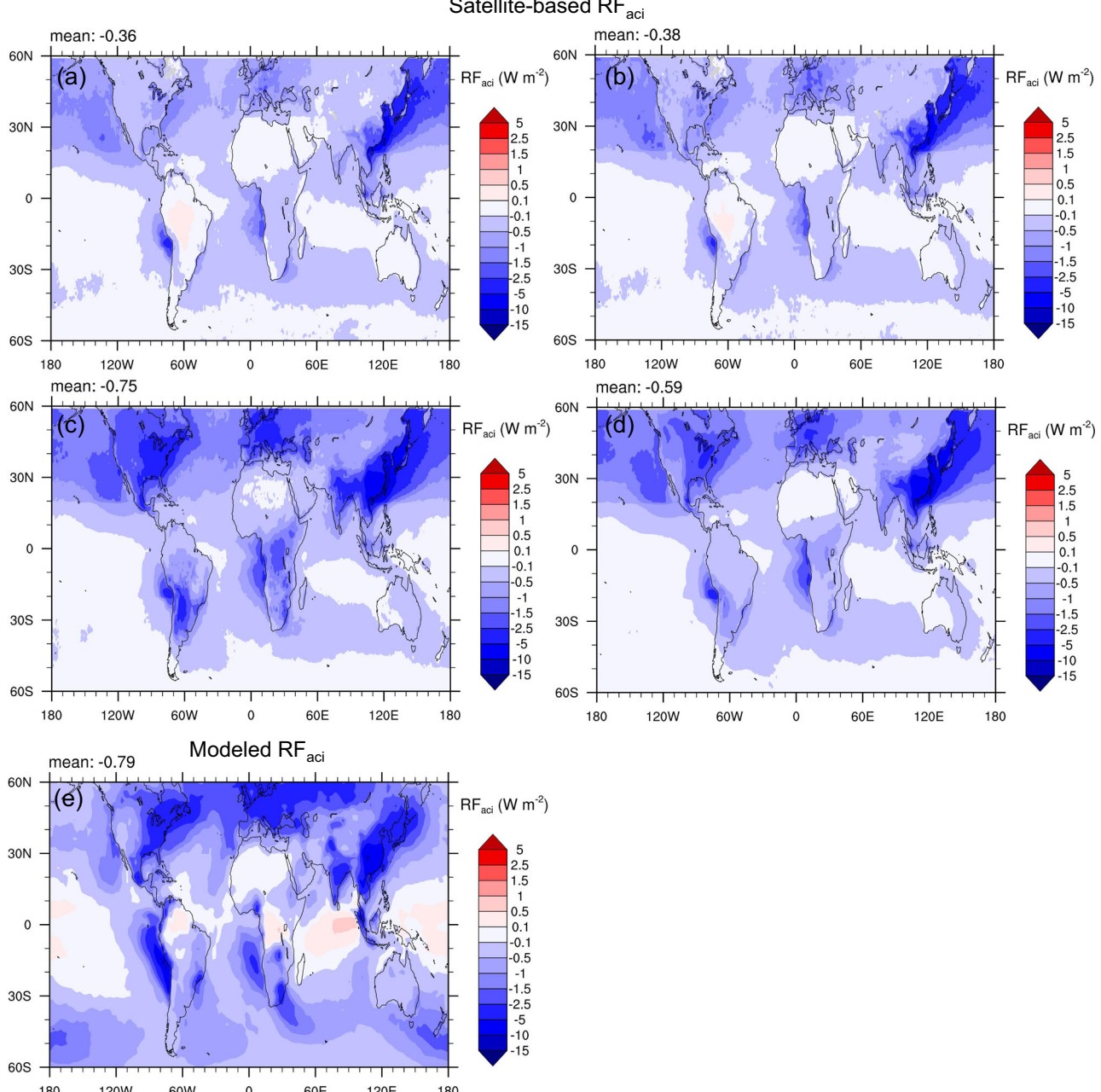

**Fig. 4 Annual mean first indirect forcing (RF$_{aci}$) at a global scale (60°S to 60°N) for different scenarios.** The RF$_{aci}$ calculated from **a** MODIS-retrieved aerosol optical depth (AOD) and CERES-retrieved cloud droplet number concentration ($N_d$) data pairs (Aero_Cld_Modis), and from MERRA-2 re-analyzed AOD and CERES-retrieved $N_d$ data pairs for **b** the scenario that aerosol and cloud retrievals are simultaneously successful (Aero_Cld, including same cloud samples with Aero_Cld_Modis), **c** the scenario that only cloud retrievals are successful (Cld), and **d** the scenario including all ambient clouds (All_Cld). **e** Modeled RF$_{aci}$ from Yu et al.[44]. The respective RF$_{aci}$ values for different regions are also listed in Supplementary Table 1.

changes of $f$ (Fig. 2) and regression slopes (Fig. 3a, b) contribute to the increase of RF$_{aci}$. To gain insight into their individual effects, we conduct two additional tests on the basis of Aero_Cld. One is referred to as Aero_Cld_R that uses the regression coefficients acquired from All_Cld but keeps other quantities the same as Aero_Cld, in order to evaluate the effect of changed regression coefficients, and the other is denoted as Aero_Cld_C that is the same as Aero_Cld but uses cloud quantities from All_Cld, for the purpose of quantifying the impact of changed $f$. As demonstrated in Supplementary Fig. 5, the increased slopes lead to a relative change of +21% in the magnitude of RF$_{aci}$ (from −0.38 W m$^{-2}$ in Aero_Cld to −0.46 W m$^{-2}$ in Aero_Cld_R). The increased $f$, in turn, amplify RF$_{aci}$ by 34% (from −0.38 W m$^{-2}$ in Aero_Cld to −0.51 W m$^{-2}$ in Aero_Cld_C) with the largest increases

over NAM (−0.55 W m$^{-2}$), EUR (−0.66 W m$^{-2}$), and ASI (−0.42 W m$^{-2}$), where high anthropogenic fractions of AOD (0.57, 0.59, and 0.59, respectively) and significant enhancements of $f$ (86%, 83%, and 70%, respectively) occur simultaneously.

A recent study relying on the combinations of ln $N_d$-AOD$_f$ relationship and radiative transfer modeling reported the best estimate of RF$_{aci}$ near −0.7 W m$^{-2}$ with the associated uncertainty range between −0.5 to −1.2 W m$^{-2}$ (see ref. [9]), which is more negative than previous satellite analyses as well as our optimized AOD-based estimate (−0.59 W m$^{-2}$ in All_Cld). Here, RF$_{aci}$ for the All_Cld scenario but replacing AOD by fine-mode AOD is thus computed for comparison with the original All_Cld result (Fig. 5c). Anthropogenic fraction is correspondingly defined via the

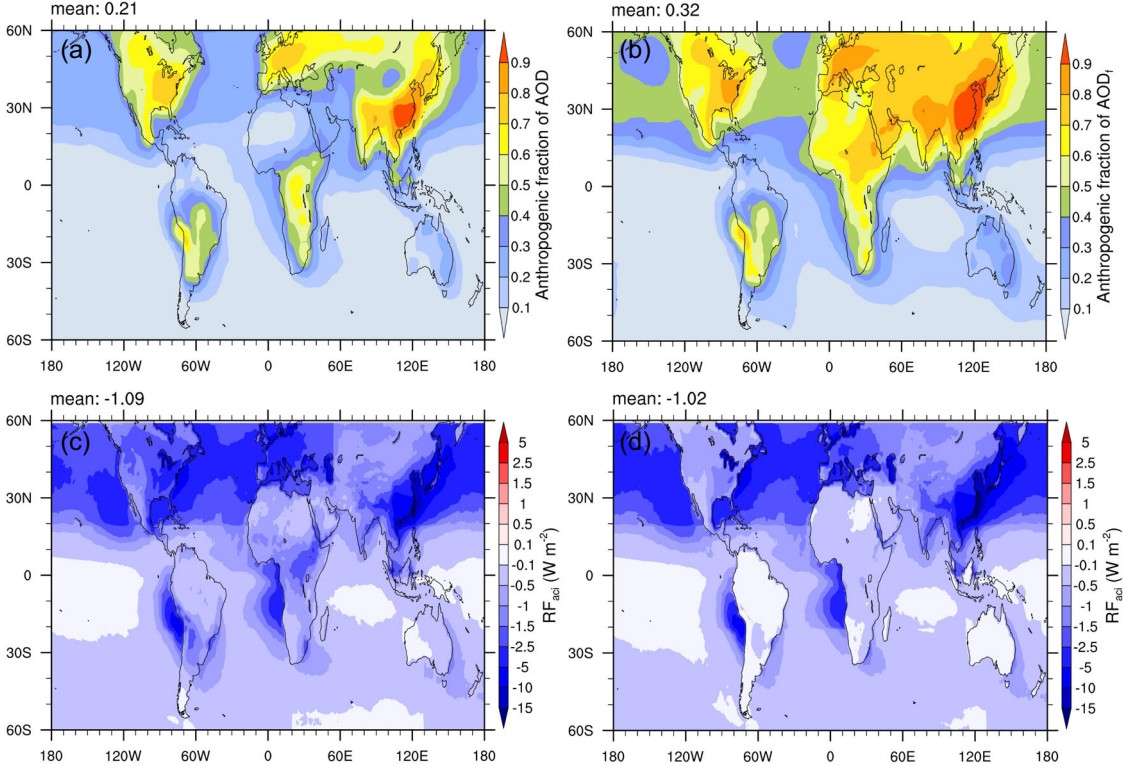

**Fig. 5 Annual (2010) averaged anthropogenic fractions and first indirect forcing (RF$_{aci}$).** Anthropogenic fractions of **a** aerosol optical depth (AOD) and **b** fine-mode AOD (AOD$_f$) from the GEOS-Chem-APM simulations. **c** RF$_{aci}$ based on MERRA-2 daily AOD$_f$ for the scenario including all ambient clouds (All_Cld). **d** RF$_{aci}$ based on monthly POLDER-3 AOD$_f$.

modeled fine-mode AOD with the same definition as MERRA-2 AOD$_f$ (see "Methods"). Since the fine-mode aerosols dominate the anthropogenic contributions, with almost the same absolute increases of AOD and AOD$_f$ from PI to PD (Supplementary Fig. 6c, d), one obtains a much larger anthropogenic fraction if AOD$_f$ rather than total AOD is applied. Figure 5a, b shows the maps of anthropogenic fractions of AOD ($f_{ant}$) and AOD$_f$ ($f_{ant-fine}$). It is clear that the most significant difference between $f_{ant}$ and $f_{ant-fine}$ occurs over dust source regions and oceans in the northern hemisphere, where coarse particles (sea salt and dust aerosols) account for a large part of total extinction so that using total AOD rather than AOD$_f$ will underestimate the anthropogenic contributions from PI to PD. When switching AOD to AOD$_f$ the estimated RF$_{aci}$ increases by 85% (from $-0.59$ to $-1.09$ W m$^{-2}$; Figs. 4d and 5c), in which the increased slope and anthropogenic fraction contribute 17% and 68%, respectively. Since a part of sea salt and dust aerosols can also serve as CCN, the actual RF$_{aci}$ should be between $-0.59$ W m$^{-2}$ and $-1.09$ W m$^{-2}$.

In addition to relying on the reanalysis product, another potential way to sidestep the sampling biases would be the use of monthly aerosol/cloud statistics from satellite observations. Figure 5d shows the RF$_{aci}$ estimated by employing monthly POLDER-3 AOD$_f$ and monthly cloud properties. The global-averaged RF$_{aci}$ ($-1.02$ W m$^{-2}$) is close to the MERRA-2 daily-based one ($-1.09$ W m$^{-2}$; Fig. 5c) but with different spatial distributions, i.e., much smaller RF$_{aci}$ over land, which is also in contrast to models (Fig. 4e). A comparison of the slopes of ln $N_d$ versus ln AOD$_f$ between the two cases shows that POLDER monthly-based slopes are significantly lower than MERRA-2 daily-based ones over land (Supplementary Fig. 7). One reason would be covariation of retrieval biases in aerosol and cloud properties, tending to underestimate aerosol–cloud correlations[17]. It should also be noted that the use of monthly statistics can only avoid the sampling bias of clouds, while aerosol information

associated with the clouds that fully cover $1° \times 1°$ grid box are impossible to be sampled. That is, this monthly AOD$_f$ would be not well representative of the number of aerosols actually linking with monthly $N_d$. Our analysis demonstrates that even though all clouds have been sampled, missing the collocated aerosols in monthly statistics can still lead to an underestimation of aerosol–cloud correlations (Supplementary Fig. 8). Comparing monthly (Supplementary Fig. 8) and daily (Fig. 3) slopes for All_Cld case, it is interesting that the former are generally larger than the latter, which might be partly due to the inclusion of short-term feedbacks or delayed responses of clouds in monthly aerosol–cloud associations.

**Important implications for a satellite-based estimate of RF$_{aci}$.** To confidently interpret past and predict future climate change, the current significant discrepancy between satellite- and model-based RF$_{aci}$ has to be reconciled. Among possible directions to fill this gap (detail in the following section), analyzing satellite results in an apples-to-apples way to model simulations is essential. Although many efforts have been made to do rigorous comparisons[39,49], e.g., selecting model outputs at satellite equatorial crossing time, and sampling cloud-top quantities in the uppermost liquid water cloud layer, etc., sampling biases discussed in this study have been largely ignored earlier.

Previous studies that deducted either aerosol–cloud correlations or radiative forcing relied on the assumptions that the clouds collected when adjacent aerosol retrievals are available are representative of all ambient clouds. Our findings here, however, demonstrated that sampling biases introduced by the inherent satellite retrieval limitation of aerosol-only cloud-free conditions systematically exclude liquid clouds with high $f$, which are predominantly thick stratiform cloud decks. These clouds, on the one hand, exert a stronger cooling effect, and, on the other hand,

were found to be more responsive to the perturbations of aerosols compared to cumuliform clouds[26]. By fixing the sampling biases, the estimated RF$_{aci}$ increases by 133% over land and 33% over the ocean, which is closer to models not only in global-averaged RF$_{aci}$ ($-0.59$ W m$^{-2}$) but also in its spatial distribution or land–ocean contrast. More importantly, the estimated magnitude of RF$_{aci}$ is almost doubled ($-1.09$ W m$^{-2}$) when replacing AOD by AOD$_f$ to derive the aerosol–cloud correlations and the anthropogenic fraction. This finding implies that previous satellite-based estimates[5,7,27,28] have substantially underestimated the RF$_{aci}$, especially over land, and further highlights the necessity of accounting for the sampling biases as well as utilizing fine-mode AOD (or other adequate CCN proxies) instead of total AOD in future satellite investigations.

Although the use of monthly aerosol/cloud retrievals can largely sidestep the sampling bias on clouds and also include short-term feedbacks or delayed responses of clouds, the problem of missing aerosol information under cloudy sky remains. This will cause a significant underestimation of RF$_{aci}$, over land where aerosols have large inter-daily variability. The result implies that the conclusion regarding spatial distribution (land–sea contrast) of RF$_{aci}$, should be drawn carefully if monthly statistics are applied.

In this study, the enhanced RF$_{aci}$, from AOD- to AOD$_f$-based estimates is mainly induced by the increased anthropogenic fraction. The anthropogenic fraction depends not only on the increase in the anthropogenic aerosols but also on the preindustrial background (Supplementary Fig. 6). To understand the sensitivity of RF$_{aci}$ to the choice of anthropogenic fraction, the RF$_{aci}$, are also computed with anthropogenic fractions of AOD$_f$ from the simulations of (1) AeroCom phase1 models[50] with the preindustrial year 1750 as a reference and AeroCom phase2 models[51] with the preindustrial year 1850 as a reference in addition to GEOS-Chem-APM modeled ones (Supplementary Table 3 and Supplementary Fig. 9). The result indicates that the RF$_{aci}$ strongly scales with the anthropogenic fraction of AOD$_f$, whereas there is no evident difference in global-averaged RF$_{aci}$ between daily- and monthly-based estimates. With the adequate CCN proxy (AOD$_f$ here), and meanwhile, sidestepping the sampling biases by relying on daily re-analyzed and monthly retrieved AOD$_f$ respectively, the RF$_{aci}$ are estimated to range from $-1.02$ to $-1.68$ W m$^{-2}$ when applying different anthropogenic fractions (Supplementary Table 3), which highlights the need for more meaningfully constrained anthropogenic fraction.

## Discussion
This study focuses on the roles of sampling biases and choice of the CCN proxy in the satellite-based estimate of RF$_{aci}$, with a clear demonstration of the potential magnitude of the impacts of both. It should, however, be noted that the exact forcing value is also affected by other potential sources of the uncertainties, which are noteworthy for future explorations.

As mentioned above, the retrievals of AOD and cloud quantities (CER/$\tau_c$) exhibit large biases in scenarios involving broken clouds, and importantly, the covariation of retrieval biases in AOD and CER (thus $N_d$) appears to underestimate the cloud albedo effect[17]. Here, we made an initial attempt to assess "overcast" clouds by restricting retrievals to clouds with $f > 80\%$. This threshold was also adopted by previous studies[16]. It is found that there is no systematic difference in aerosol–cloud correlations between the analyses based on all cloud samples and "overcast" clouds only (Supplementary Fig. 10). With the use of CERES data here, which does not have any pixel-level flag identifying overcast or partly cloudy conditions, we thus do not distinguish between overcast and broken clouds. But it is believed that using pixel-level cloud retrievals with the overcast flag (e.g.,

MODIS level-2 product) would be a useful exercise to focus on homogeneous and overcast pixels for trusted high-quality retrievals[52]. In addition, it has been demonstrated that co-variability of aerosol and precipitation induced by wet removal can confound the interpretation of aerosol–cloud–precipitation interactions[53]. However, our analysis has been restricted to low liquid clouds, with liquid water path overall lower than 100 g m$^{-2}$ (Supplementary Fig. 11), thus generating very little precipitation. Also, non-precipitating cloud pixels have been found to account for ~90% of all footprints globally[54]. Therefore, the co-variability of aerosol and precipitation is unlikely to significantly affect the results presented here.

A single slope of ln $N_d$ versus ln AOD is clearly not representative of a variety of cloud types. In situ-[26] and satellite-based[25,55] investigations have revealed that both cloud microphysical properties and vertical cloud structure have distinct responses to aerosol perturbations for stratocumulus and cumulus clouds, which highlights the importance of conducting regression analysis for each individual cloud type. Recently, Unglaub et al.[56] developed a new approach to classify cloud types at cloud scale by using a newly developed cloud-based height retrieval[57] in conjunction with cloud-top height variability, which makes it possible to obtain a cloud-type-based estimate of RF$_{aci}$ globally in future studies.

Methodological biases are partially responsible for the uncertainties of RF$_{aci}$ estimate as well. Given the large spatial variations in aerosol loading, aerosol type, cloud type, and meteorological conditions, spurious correlations between aerosol and cloud quantities would occur when analyzing satellite datasets over large regions. Grandey and Stier[58] pointed out that for regions of $60° \times 60°$, close to the scale used in our study, this methodological error might overestimate the RF$_{aci}$ by ~80% relative to that derived from temporal variability only within $1° \times 1°$ region. They also suggested that conducting statistical analysis over regions smaller than $4° \times 4°$ can greatly avoid this error. Unfortunately, the insufficient data samples prevent us from doing such analysis at this time. However, the attempts of combining satellite datasets from multiple platforms, or employing pixel-level retrievals (e.g., $1 \times 1$ km$^2$ cloud retrievals in MODIS level-2 dataset) in future studies, might be useful to minimize this error. In addition, Patel et al.[28] found that for the derivation of regression coefficients of the relationship between planetary albedo and cloud properties (see Eq. (2) in "Methods"), the nonlinear least square statistical approach can increase the correlation by 21–23% and reduces the error compared to the multilinear regression approach[5], thus reducing the uncertainty of RF$_{aci}$. This finding implies the need to re-estimate RF$_{aci}$ by employing the updated approach. For the sake of comparison with previous studies[5,7], however, the multilinear regression is still used here. Finally, it has been demonstrated that more directly retrieved CCN leads to increases in the $N_d$–aerosol slope[23]. Hasekamp et al.[23] found a strong increase in RF$_{aci}$ when using a polarimetric retrieval of column-CCN over oceans, which, in combination with the results presented here, implies a still stronger RF$_{aci}$. Beyond RF$_{aci}$, also rapid adjustments such as changes in cloud liquid water path and in $f$ add to the total radiative effect of aerosol–cloud interactions, the effective radiative forcing (ERF$_{aci}$). These adjustments approximately scale with RF$_{aci}$[10,59], and thus our result is relevant for ERF$_{aci}$, too.

## Methods
**Satellite and reanalysis data**. The satellite data used here are similar to that in previous studies using the same approach[5,7] but with an updated version, including cloud and radiative products from the CERES Single Scanner Footprint (SSF) Edition-4 dataset at a $20 \times 20$-km$^2$ resolution and Dark Target and Deep Blue combined aerosol products from the MODIS Collection 6.1 dataset at $1° \times 1°$ level-3 resolution. Both instruments are on board the Terra platform with an equatorial crossing local time at ~10.30 a.m. To determine RF$_{aci}$ induced by fine-mode

aerosols, $AOD_f$ is also required. However, aerosol size information (such as Angström exponent and fine-mode fraction) retrieved from MODIS over land may be problematic[31], hampering the derivation of $AOD_f$. For this reason, we also employ the daily $1° \times 1°$ POLDER-3 $AOD_f$ product retrieved by the GRASP algorithm, which was found to have good quantitative skill over land[41,43].

To fill the data gap caused by satellite sampling, the re-analyzed AOD and $AOD_f$ from the MERRA-2 dataset at a $0.5° \times 0.625°$ resolution, are also utilized in this study, which has been found to compare well with other independent observations from the ground, aircraft, and shipborne instruments[32,60]. Our comparison also shows a good agreement between POLDER-3 and MERRA-2 $AOD_f$ (Supplementary Fig. 12). Here, $AOD_f$ is defined as the sum of AODs of sulfate, black carbon and organic aerosol, and 30% sea salt aerosol. MERRA-2 assimilates AOD retrieved by multiple satellite sensors (AVHRR, MODIS, MISR) and the ground-based remote sensing network (AERONET) to correct for model departures from observations[32]. The re-analyzed AOD thus combines the advantages of both satellites and models, providing full spatial and temporal coverage while keeping a strong connection to observed aerosols. Hourly re-analyzed AOD/$AOD_f$ at the equatorial crossing local time of satellite is used to match to the satellite-observed parameters.

The MODIS, CERES, and reanalysis data span the time period January 2002 through December 2018, and POLDER-3 data are from March 2005 to October 2013, providing sufficient samples to obtain statistically significant results. Supplementary Table 2 summarizes the surface type, aerosol, cloud, radiative parameters required for the statistical analysis and/or the $RF_{aci}$ calculations. For the AOD/$AOD_f$, data at the resolution of $1° \times 1°$ (MODIS and POLDER-3) and $0.5° \times 0.625°$ (MERRA-2) are both projected to the higher resolution of $20 \times 20$ km$^2$ to match the SSF products.

**Methodology for calculating $RF_{aci}$ in the context of liquid water clouds.** For the calculation of $RF_{aci}$, a critical procedure is to determine the change in $N_d$ due to anthropogenic aerosols based on the relationship between $N_d$ and AOD ($AOD_f$). However, $N_d$ is not routinely retrieved in current satellite product, and needs to be empirically computed from cloud effective radius ($r_e$) and cloud optical depth ($\tau_c$) for liquid water clouds assuming adiabaticity[13] as follows:

$$N_d = \gamma \tau_c^{1/2} r_e^{-5/2} \qquad (1)$$

where $\gamma$ is an empirical constant with the value of $1.37 \times 10^{-5}$ m$^{-0.5}$ (see ref. [13]). It was suggested that the $N_d$ retrieval performs better for relatively homogeneous, optically thick and unobscured stratiform clouds under a high solar zenith angle condition[16,61–64].

Quaas et al.[5] have extended Loeb's[65] approach by accounting for the contribution of the clear part of a scene to estimate planetary albedo ($\alpha$), where $\alpha$ can be described by a sigmoidal fit as

$$\alpha \approx (1-f)\left[a_1 + a_2 \ln \tau\right] + f\left[a_3 + a_4 (f\tau_c)^{a_5}\right]^{a_6} \qquad (2)$$

where $\tau$ is AOD ($AOD_f$), and fitting parameters ($a_1 - a_6$) are obtained by a multilinear regression ($a_5$ is set as 1[7]). The performance of the multilinear regression fitting has been verified by previous studies. Their results showed that the fitted albedo overall agrees very well with both CERES-retrieved albedo[5] and the albedo simulated by a radiative transfer model[28]. Quaas et al.[5] suggested that the $RF_{aci}$ for anthropogenic aerosols can be expressed as

$$RF_{aci} = f_{liq} A(f, \tau_c) \frac{1}{3} \frac{d\ln N_d}{d\ln \tau}[\ln \tau - \ln(\tau - \tau^{ant})]S \qquad (3)$$

with $f$ the cloud fraction, $f_{liq}$ the fraction of liquid water clouds, $\tau^{ant}$ the anthropogenic AOD ($AOD_f$), and $S$ the daily mean incoming solar radiation, respectively. Here, $A(f, \tau_c) = a_4 a_5 a_6 \left[a_3 + a_4 (f\tau_c)^{a_5}\right]^{a_6-1} (f\tau_c)^{a_5}$.

Since the fitting parameters ($a_1 - a_6$) and the slope of the linear regression fit between $\ln N_d$ and $\ln \tau$ can vary both temporally and spatially, we conduct the regression analysis separately for fourteen regions (Supplementary Fig. 1) on a monthly basis. To obtain more reliable fitting parameters, only a subset of the data with smaller retrieval biases are used for statistical regressions, excluding the retrievals involving bright surfaces, high solar zenith angle (>65°), thin clouds (liquid water path, $L < 20$ g m$^{-2}$), multilayered clouds. In addition, the lowest 15% of data for AOD ($AOD_f$) are also excluded, since the slopes of $\ln N_d$ versus $\ln$ AOD ($AOD_f$) are quite sensitive to small AOD ($AOD_f$) changes, which are not well-characterized by satellites[18]. As for the calculation of $RF_{aci}$, these data are re-included in order to obtain an estimate in a more consistent manner with the model, i.e., minimizing the sampling biases.

As a key parameter in estimating $RF_{aci}$, $\tau^{ant}$ can be derived either from column-integrated aerosol properties involving size and absorption information from satellite observations[28,33,66,67] and/or reanalysis data[6], or from model simulations[7]. Total (fine-mode) $\tau^{ant}$ in our study is obtained by multiplying AOD ($AOD_f$) from MODIS/MERRA-2/POLDER-3 and anthropogenic AOD ($AOD_f$) fraction ($f_{ant}$; $f_{ant-fine}$) simulated by Ma et al.[7]. $AOD_f$ in the model is also defined in consistent with the definition for MERRA-2 $AOD_f$. To derive $f_{ant}$ ($f_{ant-fine}$), two simulations were conducted with one for PD and the other for PI (with the preindustrial year 1750 as reference) aerosol particle and aerosol precursor gas emissions by employing GEOS-Chem-APM model[44], in which an advanced multitype, multicomponent, size-resolved microphysics model was coupled to a global 3-D model of

atmospheric chemical model GEOS-Chem. More detail of the model description can be found in Ma et al.[7]. Using the same approach and $f_{ant}$ as Ma et al.[7] allows us to qualify the impact of satellite product updates. The $RF_{aci}$ in this study is calculated for the year 2010 in order to be consistent with the latest Intergovernmental Panel on Climate Change (IPCC) report[4] that used the reference year 2010 for PD conditions. In addition to the "standard" simulation mentioned above, the $f_{ant-fine}$ from the simulations of AeroCom phase1 and phase2 models (obtained from the MACv2 aerosol climatology product[68]), as well as GEOS-Chem-APM model but with $AOD_f$ defined as the sum of AODs of sulfate, black carbon and organic aerosol, are also adopted to understand the sensitivity of $RF_{aci}$, to the choice of the anthropogenic fraction.

## Data availability
All data analyzed in this study are publicly available. The CERES SSF product is available from https://opendap.larc.nasa.gov/opendap/CERES/SSF/. The MODIS Level 3 Collection 6.1 product is available at https://ladsweb.modaps.eosdis.nasa.gov/archive/allData/61/MOD08_D3/. The POLDER-3 product is available from https://www.grasp-open.com/products/polder-data-release/. The MERRA-2 reanalysis product is collected from https://goldsmr4.gesdisc.eosdis.nasa.gov/data/MERRA2/. The MACv2 aerosol climatology product is available at ftp://ftp-projects.mpimet.mpg.de/aerocom/climatology/MACv2_2018/.

## Code availability
The codes for calculations and data processing are available from the corresponding authors upon request.

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

## Acknowledgements

The authors are grateful for the ease of access to CERES, MODIS, POLDER-3, MERRA-2, and MACv2 data, provided by the NASA Langley Research Centre, the NASA Goddard Space Flight Centre, the Centre National d'Etudes Spatiales, and the NASA Global Modeling and Assimilation Office, and the Max-Planck-Institute, respectively. This research has been supported by the National Key R&D Program of China (grant no. 2019YFA0606802), the National Natural Science Foundation of China (grant nos. 42061134009 and 41975002), the Second Tibetan Plateau Scientific Expedition and Research (STEP) program (2019QZKK0103), the NASA (grant nos. 80NSSC19K1275 and NNX17AG35G), National Science Foundation of US (grant no. AGS1550816), the Federal Ministry of Education and Research in Germany (Bundesministerium für Bildung und Forschung; BMBF) through the FONA research project "CHANCE" (FKZ 01DR20001), and China Scholarship Council.

## Author contributions

H.J. and F.Y. designed the research. H.J. and X.M. performed the research. H.J. drafted the paper. X.M., F.Y., and J.Q. validated and debugged the results. All authors contributed to revising the paper.

## Competing interests

The authors declare no competing interests.
