## [Peer Review File · Nature Communications]

Reviewer comments, first round

Reviewer #2 (Remarks to the Author):

Review of "Significant underestimation of radiative forcing by aerosol–cloud interactions derived from satellite-based methods" by H. Jia et al.

The authors present a compelling case for missing aerosol forcing in satellite based estimates due to sampling biases introduced when seeking to match cloud properties with (clear-sky) aerosol estimates. The authors decompose this bias into a component due to a mis-estimation of the regression slope, and a component due to a mis-sampling of the underlying cloud property distributions. They seek to account for these biases using an assimilated aerosol product which is available under all-sky conditions. They examine potential deficiencies in this assimilated product and provide an updated satellite based estimate of RF_{aci}.

The paper addresses an important issue and is generally well structured. I am happy to recommend publication once the following points have been satisfactorily addressed.

Major concerns:

- Using an assimilated aerosol product provides a clear demonstration of the potential magnitude of the affect of the sampling biases discussed here, but the exact forcing values will depend on the details of the model, and in particular on the co-variability of aerosol and precipitation. This was most clearly demonstrated in Gryspeerd et al. 2015 (<https://doi.org/10.5194/acp-15-7557-2015>), who showed the difficulty in representing the correlation between AOD and cloud properties in both observations and global models. How does MERRA-2 represent activation and wet-removal, how has this been validated and what affect could it have on the results presented here?
- Further, while the authors have demonstrated that the MERRA-2 aerosol product does a reasonable job of reproducing MODIS AOD, it would be valuable to present (or cite a) validation against an independent instrument, especially given the importance of this to the manuscript.
- Also, the manuscript can be difficult to read at times and would benefit from editing help from someone with full professional proficiency in English.

Minor issues:

- L25 - '...with more droplets.' This is only true for a fixed amount of liquid water.
- L67 - Suggest replacing 'data-tied' with 'consistent'
- L108 - 'have' -> has
- L108 - Remove 'pretty' colloquialism
- L110 - 'ones' -> 'clouds'
- L207-208 - "Moreover, although... " This sentence compares RF_{aci} to RF_{aci} and doesn't seem to make sense.

Reviewer #3 (Remarks to the Author):

The authors put forward a straight-forward, but impactful estimate of the RF_{aci}. By neglecting clouds with high coverage and optical depth we underestimate how strong RF_{aci} is. The authors correct this and move toward reconciling literature estimates of RF_{aci}. While aerosol column brightness based estimates have many well known problems, it is important to characterize why older estimates of RF_{aci} tend to be weak compared to newer estimates and GCMs.

While I feel this study may merit publication in NCOMMS I think the addition of a GCM-only analysis to support their observational analysis following Ma et al. [2018] is needed to make sure it is robust. Their analysis can be replicated by sampling data from a GCM running a satellite simulator easily.

The authors neglect a lot of the existing literature on how to appropriately retrieve Nd [Daniel P. Grosvenor et al., 2018] and utilize an unusual retrieval methodology (using CERES observations). Their decision to use this data set instead of a more well-established one must be discussed.

Major:

P2 L29: I think the distinction needs to be made that these are estimates based on covariability between column measurements of aerosol and cloud properties. Estimates based on metrics like aerosol mass tend to have higher values [Bellouin et al., 2020; Boucher and Lohmann, 1995; McCoy et al., 2018].

P2 L36: Authors need to cite and discuss Ma et al. [2018], which shares a lot of the same techniques and goals as this paper but focusses on aerosol-cloud adjustments.

P2 Ln 60: the argument is unclear here and appears to conflate adjustments and R_{Faci}. It seems like all the authors are saying is that $d\ln Nd/d\ln AOD = f_n(CF, \tau_{cloud})$. Is the second argument that clouds with a larger cf or lower tau would tend to have a larger radiative response to changes in Nd? This is accounted for in many studies estimating R_{Faci}.

L36 the authors here and elsewhere suggest that higher tau clouds would have a larger R_{Faci}. The opposite should be true. When clouds have a low tau you get more bang for your buck for a set increase in tau (see fig. 3 of Carslaw et al. [2013]). It might be useful to show the analysis Fig. S6 into a changed tau and changed CF contribution.

Ln73: Why use CERES for cloud properties? This is highly unusual and there has no validation of the Nd from CERES. MODIS Nd could be used, which would match the MODIS AOD being used, and which has been extensively validated and discussed elsewhere [Daniel P. Grosvenor et al., 2018]. On Ln 251 the CERES data being used is discussed, but it is unclear what data is removed, such as high SZA, or low CF [D. P. Grosvenor and Wood, 2014]. This has a substantial impact on the study and needs to be clearly stated.

Ln 109 cite Hamilton et al. [2014]

Ln 221 I believe the authors are conflating systematic bias and random uncertainty in their citation and discussion. Random uncertainty in Nd was reported at 80% on a pixel level in Daniel P. Grosvenor et al. [2018] and this dropped substantially when averaged to a 1x1° region. However, as stated in that review paper the systematic bias in the Nd relative to in situ was low. The random uncertainty discussed here does not affect the results of the present publication beyond increasing uncertainty in their regression coefficients.

Minor:

P1 L12: I think that the authors mean the uncertainty due to differing estimates of R_{Faci} from GCMs and obs.

P1 L17: Increases in magnitude or becomes more positive?

P2 L28 'in'

Bellouin, N., et al. (2020), Bounding Global Aerosol Radiative Forcing of Climate Change, Rev. Geophys., 58(1), doi: 10.1029/2019rg000660.

Boucher, O., and U. Lohmann (1995), The sulfate-CCN-cloud albedo effect, Tellus B, 47(3), 281-

300, doi: 10.1034/j.1600-0889.47.issue3.1.x.

Carslaw, K. S., et al. (2013), Large contribution of natural aerosols to uncertainty in indirect forcing, *Nature*, 503(7474), 67-71, doi: 10.1038/nature12674.

Grosvenor, D. P., and R. Wood (2014), The effect of solar zenith angle on MODIS cloud optical and microphysical retrievals within marine liquid water clouds, *Atmos. Chem. Phys.*, 14(14), 7291-7321, doi: 10.5194/acp-14-7291-2014.

Grosvenor, D. P., et al. (2018), Remote Sensing of Droplet Number Concentration in Warm Clouds: A Review of the Current State of Knowledge and Perspectives, *Rev. Geophys.*, 56(2), 409-453, doi: 10.1029/2017rg000593.

Hamilton, D. S., L. A. Lee, K. J. Pringle, C. L. Reddington, D. V. Spracklen, and K. S. Carslaw (2014), Occurrence of pristine aerosol environments on a polluted planet, *Proceedings of the National Academy of Sciences*, 111(52), 18466-18471, doi: 10.1073/pnas.1415440111.

Ma, P.-L., P. J. Rasch, H. Chepfer, D. M. Winker, and S. J. Ghan (2018), Observational constraint on cloud susceptibility weakened by aerosol retrieval limitations, *Nat. Commun.*, 9(1), 2640, doi: 10.1038/s41467-018-05028-4.

McCoy, D. T., F. A. M. Bender, D. P. Grosvenor, J. K. Mohrmann, D. L. Hartmann, R. Wood, and P. R. Field (2018), Predicting decadal trends in cloud droplet number concentration using reanalysis and satellite data, *Atmos. Chem. Phys.*, 18(3), 2035-2047, doi: 10.5194/acp-18-2035-2018.

REVIEWER COMMENTS

Thanks to the reviewers for their constructive comments and very helpful suggestions, which have allowed us to clarify and improve the manuscript. Below we address the reviewers' comments, with the reviewer comments in black, and our responses in blue. We have revised the manuscript accordingly.

Reviewer #1 (Remarks to the Author):

Review of paper:

Significant underestimation of radiative forcing by aerosol-cloud interactions derived from satellite based methods by H. Jia et al.

Positives

- timely topic
- efforts to constrain modeling freedom with observational constrains
- presentation of globally ACI patterns

Concerns

- use of AOD rather than AOD_f (how are dust events excluded?)
- poor description on obs and model interpret. sources (years, temp.resolution) until ch4
- only a simple formula with many regional fit parameters to estimate ACI

We thank the reviewer for taking the time to assess the manuscript and for providing helpful comments and suggestions to improve the manuscript. We have revised the manuscript carefully according to the reviewer's comments. Please see the following detailed point-by-point responses.

General comments

The paper speculates on the cooling impact by anthropogenic aerosol through modifications to (low altitude) clouds. This indirect climate cooling impact globally dominates the direct cooling by the added anthropogenic aerosol presence. Thus, constraining the indirect aerosol is a big step towards improving climate assessments and predictions (by reducing the aerosol associated large climate sensitivity).

The paper employs satellite observations but it also involves model interpretations in efforts to explain why purely model based estimates usually suggest stronger (climate cooling) indirect aerosol effects than those associated with satellite

observations. It is argued, that this difference should be attributed to an improper observational approach over continents for two reasons:

- Observation constrained aerosol indirect effects are likely underestimated over land, as cases of (aerosol vs cloud) number associations over land (on a daily basis though) are missing (as only overcast cloud elements [then no nearby AOD data] are reliable for quality retrievals of microphysical cloud properties)

- MODIS AOD data over land are apparently biased high (compared to data assimilations with MERRA).

When substituting MERRA AOD data over land (where MODIS AOD retrievals are not possible) then it is shown that with the added extra cases over land the cloud fraction and cloud optical depth is increased, so that with a reduced MERRA AOD (and a lower natural base state) the indirect effect sensitivity is increased. (I assume that is the thinking here..?)

The analysis below (see the response to Line 83 for more details) shows that there is no systematic difference in aerosol-cloud correlations between the analyses based on all cloud samples and 'overcast' clouds only. Thus, the inclusion of more reliable cloud retrievals would not be responsible for the larger RF_{aci} .

The sampling biases artificially and systematically exclude clouds with high cloud fraction (f) and cloud optical depth (τ_c), which are predominantly thick stratiform cloud decks.

- On the one hand, stratiform clouds have been found to be more sensitive to the perturbations of aerosols compared to cumulus clouds (Jia et al., 2019), so missing these clouds will underestimate the regression slope of $\ln Nd$ versus $\ln AOD$, thus RF_{aci} .
- On the other hand, clouds with high f and τ_c exert a stronger cooling effect by reflecting more incoming shortwave radiation than thin and broken cumulus clouds. Missing these clouds in the calculation of RF_{aci} (see equation (3) in Methods), will also lead to a lower RF_{aci} .

The effect of retrieval artifact alone was investigated by comparing two tests with the same cloud samples but different AOD products (MODIS vs. MERRA-2), i.e., not involving sampling biases. It is demonstrated that the biased high retrieved-AOD within cloudy pixels also partly contributed to the underestimation of RF_{aci} by satellite-based approach.

The new ACI results, which (unfortunately) are estimated via a simple relationship with many fit parameters (rather than comprehensive radiative transfer) show now strong aerosol indirect effects near anthropogenic sources over continents – usually even stronger than over outflow regions over oceans. This is a bit puzzling because I would have expected largest contributions over oceans due to their dark surface and the cleaner background. Also with lower solar insolation at higher (northern) latitudes, large ACI values over land are a bit puzzling.

We thank the reviewer for this comment. The result that RF_{aci} over land is larger than even over adjacent oceans indeed is the surprising, and new, result of our study. The reviewer highlights two arguments: the role of the surface albedo and the role of the solar zenith angle. Although no radiative transfer model was used, the influence of surface albedo on the estimate of RF_{aci} has been taken into account in our study. We adopted a sigmoidal fit over each continental and oceanic region separately to predict the planetary albedo through aerosol and cloud observations, so the information regarding surface albedo (bright land and dark ocean) was actually included in the regression relationship. Also the solar zenith angle is properly taken into account in the computations via the geographically-resolved top-of-atmosphere incident radiation. The key reason why RF_{aci} is largest over land is the much larger anthropogenic perturbation of the AOD.

The stronger aerosol indirect effect over continents is mainly due to the sharp increase of anthropogenic AOD fraction (due to the emission difference from preindustrial (PI) to present-day (PD)), whereas the increase is relatively weaker over oceans, even for outflow regions over oceans (Figures 4a&b). The enhanced amount of aerosol leads to a significant increase in Nd (ΔN) over land, hence resulting in a stronger forcing.

To illustrate this, we compare the anthropogenic fraction and solar insolation over Europe (R1, high anthropogenic fraction but small solar insolation) and Southeast Atlantic (R2, low anthropogenic fraction but large solar insolation). This example illustrates that from R2 to R1, the anthropogenic fraction increases by almost 300 %, while the solar insolation only decreases by 17 % (Figure R1).

Figure R1. Annual (2010) averaged anthropogenic fraction of AOD from the GEOS-Chem-APM simulations (left), and TOA solar insolation flux ($W m^{-2}$; right) from the CERES product. Regions denoted as R1 and R2 represent Europe and Southeast Atlantic, respectively.

These results (I like the presentations in global maps and regional averages) should be understood in a relative sense (so I do not go into numbers) as the anthropogenic definition is critical in any ACI estimate because this definition automatically also sets the pre-existing background reference. In that sense (with the need to associate retrieved cloud droplet concentrations the aerosol concentrations) it is strongly recommended to work with fine-mode AOD (rather than with total AOD) and also define anthropogenic AOD via an anthropogenic finemode fractions (from global modeling).

Thanks for the suggestion. This is a key change in the revision that impacted the results. As suggested by the reviewer, we have now in the revision also adopted the anthropogenic fine-mode AOD (AOD_f), which is more representative of aerosol number, to estimate the RF_{aci} (Figure 4d). The results indicate that the estimated RF_{aci} is greatly enhanced (-1.22 W m^{-2}) compared with RF_{aci} calculated from AOD. This is mainly due to the increased anthropogenic fraction over oceans and dust source regions in the northern hemisphere (Figure 4c). Over these regions, the coarse particles account for a large part of total extinction so that using total AOD rather than AOD_f will underestimate the anthropogenic contributions from PI to PD. The associated discussions have been inserted in the revised manuscript (line 153-167; line 209-221).

ACI relevant aerosol and cloud retrievals over continents are admittedly difficult and I would start with more confident relationship over oceans and then apply those over continents (such an approach though did not yield the enhanced ACI of this paper over continents – as background aerosol near sources was higher and surface reflectance is usually higher over land than oceans).

The aerosol and cloud retrievals have been extensively employed to investigate aerosol-cloud correlations as well as indirect radiative forcing over continents by many previous studies (Bréon et al., 2002; Quaas et al., 2008; Grandey & Stier, 2010; Ma et al., 2014), which generally has the ability to quantify aerosol-cloud associations although some retrieval artifacts would occur (Várnai & Marshak, 2009; Zhang et al., 2005). A comprehensive exploration of retrieval biases is beyond the scope of this paper, and the relevant references have been cited in the main text.

The enhanced ACI (RF_{aci}) over continents is caused by the sampling bias, the focus of our study. When clouds fully cover the grid boxes used for retrieval, hampering any AOD retrievals, these aerosol-cloud data pairs were not sampled for analysis in previous studies, then the sampling bias occurred. It can influence the estimate of RF_{aci} by not only changing the regression slope of $\ln Nd$ versus $\ln AOD$ but also altering cloud fraction (f) and cloud optical depth (τ_c) directly, key parameters in the calculation of RF_{aci} (see equation (3) in Methods). It is illustrated in Figure 1 and Table 1 that the sampling bias is much more serious over land than ocean. Furthermore, due to large differences in aerosol chemical components, dynamic, and thermodynamic conditions, the aerosol-cloud correlations also differ significantly between continental and oceanic regions (Grandey and Stier, 2010). For those reasons, it might be inadequate to apply the results over oceans to continents as mentioned by the reviewer.

Another point (also mentioned in the discussion) is that ACI will also depend on the cloud type, and that ACI investigation by cloud-type would be a nice exercise to demonstrate, if such stratification matters or could be ignored.

We agree that a cloud-type-based analysis is useful for a more accurate estimate of RF_{aci} , which has also been discussed in detail in the main text. However, it is difficult to classify cloud types accurately with the passive remote sensing data used in this study. Due to strict revision time, we will investigate this a bit later by combining the active remote sensing, as done by Unglaub et al. (2020).

Also (with relative small impact in number associations between aerosol and cloud between daily and monthly statistics) an analysis of monthly average should be considered to offer more direct aerosol and cloud property associations over land (without a reliance on data assimilations) though at much reduced statistics.

Thanks for the suggestion. As a test, we compared the regression slopes of $\ln N_d$ versus $\ln AOD$ computed from daily and monthly statistics based on purely satellite data (Figure R2). Although the monthly-based slopes are expected to be larger due to the inclusion of clouds with higher CF compared with daily-based ones (as illustrated by comparing *All_Cld* with *Aero_Cld* slopes), the results in Figure R2 do not show a systematic difference between the two. This could be attributed to

- the analysis of monthly average still suffers from to some extent the problem of missing data , especially for high latitude areas in the Northern Hemisphere in winter (Figure R3), where MODIS monthly AOD retrievals are generally not available.
- using monthly mean aerosol and cloud property statistics to conduct the linear regression strongly reduces the variability in input parameters compared with a method using instantaneous daily satellite data, and thus underestimates the slopes of $\ln N_d$ versus $\ln AOD$ (Quaas et al., 2011).
- for the linear regression over a large region with meteorological fields varying significantly, the slope derived from monthly mean statistics tends to suffer more from the spatial covariation of aerosol and meteorological fields compared with instantaneous satellite observations.
- also, as stated by the reviewer, much reduced samples for monthly-based slopes will introduce additional uncertainties in the regression analysis.

Therefore, in order to avoid data missing as well as maintain a reliable regression analysis, adopting data assimilation would be a more optimized method than using monthly average satellite observations.

Figure R2. Annual averaged slopes of the linear regression $\ln N_d$ versus $\ln AOD$ based on daily (red) and monthly (blue) statistics from satellite observations. Annual averaged slope here is the average of the monthly slopes. The standard deviation of the inter-monthly variability of the regression slopes is shown as an error bar.

Figure R3. Ratio of the number of missed cloud samples in the analysis of monthly average to the total number of cloud samples in December-January-February.

When reading the contribution I was dismayed to start guessing through the paper on data and method detail until I got to chapter 4. If chapter 4 has to stay in the order (I personally would move it after the introduction) than at least outline the major elements and refer to chapter 4, before introducing results. Also some infos provided in the supplementary section would do well in the main paper, if space allows.

We agree that it is better to introduce data and methods in advance, unfortunately according to NCOMMS format, the *Methods* section has to be placed last. As suggested by the reviewer, we have added a brief description on the data and method at the end of the introduction and referred to chapter 4 for more details. We also referred to the *Methods* section in the main text where it's needed. As suggested, the modeled anthropogenic fraction of AOD (Figure. S2 in the supplementary section) has been moved to the main paper (Figure 4) for better understanding the RF_{aci} pattern.

Overall, the study is interesting and involves a lot of data massaging. However, conceptual elements (e.g. why with increases to cloud fraction and optical depth to ACI sensitivity increases) and methods (e.g. use AODf as proxy for aerosol number) could (should?) be improved.

According to the reviewer's suggestion, the AOD_f-based analysis has been included in the revised manuscript. As discussed in the text, both in-situ (Vogelmann et al., 2012; Jia et al., 2019) and satellite observations (Gryspeerdt and Stier, 2012) have demonstrated that the response of Nd to aerosols is more sensitive for stratiform clouds than cumulus clouds, thus the increased ACI sensitivity with cloud fraction and optical depth would result from the transition from cumulus to stratiform clouds. The importance of cloud-type-based analysis for a more accurate estimate of RF_{aci} was highlighted in the discussion section. However, it is difficult to classify cloud types accurately relying on the passive remote sensing data used in this study. Due to strict revision time, we will investigate this a bit later by combining the active remote sensing.

Minor comments

Line 29 I would argue that there are observational approaches with more negative ACI effects. For instance, based on an admittedly simple relationship between anthropogenic finemode AOD (of MACv2, Kinne TellusB 2019) with co-located CDNC of

overcast clouds over oceans, once applied to the entire globe (including land) leads to an 'observation-based' best estimate of ACI near -0.7W/m^2 (Kinne ACP, 2019). The associated uncertainty though remains large (at least between -0.5 to -1.2 W/m^2) mainly as anthropogenic fine-mode contributions are not well known.

Thanks for the comment. The value of ACI and its uncertainty range reported by Kinne (2019) has been included in the observational range of the first indirect forcing (line 30), and also discussed in the section 2.2 (line 209-211).

Line 41-42 I agree that cloud microphysical retrievals should not involve data from broken cloud fields (biases) and I strongly suggest to use fine-mode AOD (rather than total AOD) for a better representation of aerosol particle number in the atmosphere. Still then, a specific altitude distribution cannot be considered and low supersaturations need to be assumed (high supersaturations involve smaller optically not detectable aerosol sizes)

Thanks for suggestions. The analysis based on fine mode AOD has been included in the revised manuscript (as responded above). In this study, clouds with liquid water path less than 20 g m^{-2} have been excluded to avoid retrievals on optically thin and, to the extent possible, broken clouds. Furthermore, we also make an initial attempt to ensure 'overcast' clouds by restricting retrievals to clouds with cloud fractions $> 80\%$ over a $20\text{km} \times 20\text{km}$ region, and found there is no systematic difference in aerosol-cloud correlations between the analyses based on all cloud samples and 'overcast' clouds only (see the response to **Line 83** for more details).

As to the altitude distributions and supersaturations, there are good points but the data required to address these are presently not available in passive remote sensing. Active satellite retrievals will be considered for analysis in the future. As discussed by e.g. Quaas et al., (ACPD 2020), one would expect that if aerosol information was available at cloud base altitude, an even stronger statistical relationship would be expected. A statement on this is added to the revised manuscript (Line 48-49).

Line 50-53 A horizontal 1×1 grid data resolution is adopted, but what temporal resolution is used? Using daily data there is often a need to move to larger grid sizes for matches with overcast clouds ... but monthly averages would continue to allow matches on a 1×1 grid basis between quality aerosol and cloud droplet proxies.

Thanks for the reminder. Here we meant the daily resolution, which has been clarified in the revised manuscript (line 56).

Line 73 It is interesting that CERES (and not MODIS) cloud data are used.

The CERES product provides strictly collocated cloud properties and TOA radiative fluxes so that we can obtain more reliable fitting parameters of the multilinear regression that links radiation with cloud properties, and thus a more accurate estimate of RF_{aci} . For pixel-level MODIS data, however, there is a need to interpolate CERES radiative observations to MODIS resolution ($1 \times 1\text{ km}^2$), which would introduce additional uncertainties.

Also, previous satellite-based studies on the subject have made use of CERES cloud properties to estimate RF_{aci} (Quaas et al., 2008; Bellouin et al., 2013; Ma et al., 2014). Therefore, using CERES data here helps to compare with previous results.

Line 77 would you expect AOD to be so different in (humidification +, removal -) and outside clouds. And if so, do you trust the interpreting model? (now not an observation anymore)

The humidification effect exists not only in clouds but also in their nearby clear area as cloud formation requires a high humidity environment, which is thus expected to contribute little to the difference between AODs. As for the wet removal, AOD in and outside clouds would differ significantly only in the presence of precipitation. However, this study has been restricted to low liquid clouds. As indicated in Figure S8, liquid water path is overall lower than 100 g m^{-2} , thus generating very little rain. O'Dell et al. (2008) also found that non-precipitating cloud pixels account for roughly 90% of all footprints globally. Therefore, the wet removal would also not induce a significant difference between AODs in and outside clouds, which has been discussed in the revised manuscript (line 272-275).

Line 83 better define cloud-fraction (what is the threshold) and cloud optical depth (is there a limitation to overcast only low altitude clouds?). I assume you talk about (warmer) lower altitude water cloud (what if high clouds interfere?) and I hope that optical depth retrievals are only applied to overcast pixels (to avoid biases).

Thanks for the comment. As mentioned at the end of the introduction and detailed in the *Methods* section, cloud fraction is defined on a scale of $20 \times 20 \text{ km}^2$, ranging from 0 to 1. To ensure high-quality retrievals, we restricted cloud retrievals to liquid phase and single-layer clouds. Additionally, clouds with liquid water path less than 20 g m^{-2} are also excluded to avoid retrievals on optically thin and, to the extent possible, broken clouds.

In spite of this, some of broken clouds could still be included. Thus, we make an initial attempt to ensure 'overcast' clouds by restricting retrievals to clouds with cloud fractions $> 80 \%$ over a $20 \text{ km} \times 20 \text{ km}$ region. This threshold was also adopted by previous studies (Bai et al., 2020; Grosvenor et al., 2018). As demonstrated in Figure R4 (also Figure S7), there is no systematic difference in aerosol-cloud correlations between the analyses based on all cloud samples and 'overcast' clouds only. Therefore, with the use of CERES data, which do not have any pixel-level flag identifying overcast or partly cloudy retrievals, we do not distinguish overcast and broken clouds in this study. But we believe that adopting pixel-level cloud retrievals with the overcast flag (such as MODIS level-2 $1 \times 1 \text{ km}^2$ product) would be a nice exercise to demonstrate whether eliminating broken clouds matters a lot to aerosol-cloud correlations. This has been discussed in section 3 (line 259-265).

Figure R4. Annual averaged slopes of the linear regression $\ln N_d$ versus $\ln AOD$ (MERRA-2). Red and blue bars represent the results based on all cloud samples and overcast clouds only, respectively. Here, the overcast pixel is defined as a pixel with $20 \times 20 \text{ km}^2$ resolved cloud fraction larger than 80 %. The annual averaged slope here is the average of the monthly slopes. The standard deviation of the inter-monthly variability of the regression slopes is shown as an error bar.

Line 85 the description of the case is not quite clear (what scale means 'adjacent'? or does it mean 'colocated' in a 1×1 deg box?)

Thanks for the reminder. Yes, it does mean 'colocated' in a $1^\circ \times 1^\circ$ grid-box, and has been clarified in the Table 1.

Line 90 There are quite a lot of samples. Is it still based on 1×1 deg resolution (Microphysical on the region are normally retrieval at smaller scales and at overcast pixels)? What is the temporal resolution (twice daily with MODIS ... or 6 hourly). How many observational years are included?

The spatial (temporal) resolutions of cloud retrievals, aerosol retrievals, and aerosol re-analysis are $20 \times 20 \text{ km}^2$ (daily), $1^\circ \times 1^\circ$ (daily), and $0.625^\circ \times 0.625^\circ$ (hourly), respectively. Our strategy is to project coarse-resolved aerosol data to the higher resolution of cloud observations, and then select the hourly re-analysis at the equatorial crossing local time of Terra satellite to match the satellite observations. It can provide numerous instantaneous (i.e., "daily" defined for satellite observations) $20 \times 20 \text{ km}^2$ resolved aerosol-cloud data pairs for statistical analysis. A total of 17 observational years (2002 - 2018) are included. These have been detailed in the *Methods* section, and also mentioned in the main text (line 106; line 108).

Line 97 you talk about (and present in Figure 1) increases to cloud fraction and optical depth and including all cloud-retrievals – even without a successful corresponding aerosol retrieval. But a higher cloud fraction of optical depth then mostly over land does not mean that the anthropogenic aerosol associated impact is larger. If there are already sufficient CCN a few added extra ones have little impact. Similarly at larger cloud optical depth the susceptibility for albedo change is reduced. The only indirect aspect I see that with a more humid environment (of clouds) the probability of a cloud-lifetime effect (in addition to the Twomey effect) is increased.

Our analysis indicates that the missed clouds have high cloud fraction and cloud optical depth, which are predominantly thick stratiform cloud decks. On the one hand, these clouds exert a stronger cooling effect by reflecting more incoming shortwave radiation than thin and broken cumulus clouds. On the other hand, they are found to be more sensitive to the perturbations of aerosols compared to cumulus clouds (Jia et al., 2019). When the significant increases of cloud fraction and optical depth (after including all clouds) overlapping the high anthropogenic fraction over land, RF_{aci} will greatly enhance as shown in Figures 3b&d. Here, higher anthropogenic fraction means larger increase of aerosol amount from PI to PD. We agree with the reviewer that a saturation effect occurs as aerosols keep rising under a polluted background, but in this case the backgrounds for both land and ocean are quite clean under the PI period, more additional aerosols over land will consequently lead to larger ΔN , hence stronger Twomey effect. It is also to be noted that relative changes are investigated (i.e. changes in N_d with respect to \ln AOD changes), not absolute changes, in order to account for the logarithmic (saturating) behavior of the relation.

The cloud-lifetime effect describes the responses of cloud fraction and liquid water path to aerosol perturbations, but here the increased cloud fraction is a result of fixing sampling biases, not relevant to any aerosol effect. The reviewer raises an important point about cloud lifetime effects (or rapid adjustments, more generally). This will be assessed in subsequent studies.

Line 114 AI (if not AOD_f) should be used rather than AOD (also in Figure 2). You want to eliminate AOD contributed by few large (e.g. mineral dust) sizes

Thanks for the suggestion. The regression slope of $\ln N_d$ and $\ln AOD_f$ has been added to the Figure 2. The result shows that there is no systematic difference between AOD- and AOD_f-based slopes, except for dust source and outflow regions (i.e., AFR, NAO and TAO), where the slopes of $\ln N_d - \ln AOD_f$ are higher due to eliminating the influence of dust events. The associated discussions have been included to section 2.1 (line 153-167).

Line 129 explain in what way MODIS retrievals over land underestimate AIE. I assume that with a larger (MODIS) AOD the relative defined anthropogenic AOD change is larger. But, have you considered to work with fine-mode (AOD_f) or even qualitatively better (but sparser) MISR (AOD_f) statistics over land, if you do not trust MODIS over land?

The covariation of retrieval biases in aerosol and cloud properties could incur a false correlation between them, thereby underestimating cloud albedo effect, which has been mentioned in the main text, and referred to Jia et al. (2019). Specifically, AOD could be overestimated due to either cloud contaminations where spurious clouds might be present in pixels that are erroneously identified as completely clear pixels (Zhang et al., 2005), or cloud adjacency effect where cloud-free pixels are brightened by reflected light from surrounding clouds (Várnai and Marshak, 2009). Moreover, cloud retrievals applied to partially cloudy and 3-D shaped clouds pixels are expected to deviate from the retrieval assumptions of overcast homogenous cloud and 1D plane-parallel radiative transfer, and tend to result in an overestimation of cloud effective radius (CER) (Grosvenor et al., 2018).

Therefore, the covariation of biases in CER and AOD could partly offset the decrease of CER caused by increasing AOD, thus underestimate AIE.

As suggested by the reviewer, we have also used retrieved fine-mode AOD to conduct analyses, but mainly for the purpose of excluding the influence of dust events. With respect to the retrieval bias, all satellite-retrieved AOD or AOD_f would suffer from the cloud contaminations and cloud adjacency effect to some extent. However, for re-analysed AOD, it is generated by an on-line aerosol chemistry, radiation and transport model as well as assimilates less biased AOD (cloud fraction less than 70 %) from MODIS, so that it can effectively avoid the spuriously high AOD near clouds caused by retrieval problem. Thus, the re-analysed AOD is also employed to alleviate the underestimation of AIE induced by retrieval biases.

Also explain what MODIS data (deep blue, dark target, or mixed) and version (6.0 or 6.1) is used in the analysis? (Actually I found it in the supplement later but a mentioning in the text is missed.

Thanks for the reminder. MODIS Collection 6.1 Dark Target and Deep Blue combined aerosol product is used in this study, which has been clarified in the *Methods* section (line 311-312).

Line 140 I am not convinced that MERRA2 AOD interpretations under cloud skies are better – although it is possible that MODIS overestimates AOD over land.

We agree that the re-analysed AOD could also be biased due to the uncertainties of model parameterization schemes and assimilation approach (Randles et al., 2017), but the bias is expected to be systematic. However, for satellite-retrieved AOD under cloudy skies, the AOD bias could be strongly correlated with the CER bias, leading to an underestimated AIE (Jia et al., 2019). This is why the re-analysed AOD is a better option for the AIE analysis.

Line 147 ACI requires assumptions to anthropogenic AOD (which is presented in the supplement, but without much detail). This is also important as the ACI strengths depends on natural background and hereby less on (total) AOD than to aerosol number (approximated by AOD_f and AI)

Thanks for the comment. The spatial distribution of anthropogenic fraction has been moved from the supplement to the main text (Figure 4a). As responded above, we have already added the estimate of RF_{aci} based on fine-mode AOD to the revised manuscript (Figure 4d).

Line 160 Yu shows regions with positive ACI (is this due to dominant entrainment evaporation?). I also would present the data in the same order as in Table 1. The caption mention the ‘first indirect effect’ ... is this correct?

As in shown in Yu et al. (2013), the limited regions with positive ACI correspond well with regions of reduced N_d (from PI to PD) as a result of long range transport as well as the dynamic interactions among particles and precursors. As suggested, the order of tests in Table 1 has been changed accordingly. In this study, cloud lifetime effect (the second

indirect effect) has been excluded, and the RF_{aci} is only contributed by changed Nd due to anthropogenic emissions. Thus, we called it the first indirect effect.

Line 192 I hope at least there is the focus on (single layer) low altitude clouds in this study.

As stated in the *Methods* section (line 355-357), this study is restricted to liquid phase and single-layer clouds, most of which are consequently low altitude clouds.

Line 196 Good point (an alternate approach was a reliance on monthly relationships over involving trusted cloud-overcast data over oceans only). Broken clouds are also more like to reduce the Twomey (first indirect) effect by entrainment of dryer air (leading to partial cloud evaporation).

Thanks for the suggestion. We agree that the analysis of the monthly average can largely avoid the problem of missing data, but it also introduces extra uncertainties (also mentioned above), such as

- using monthly mean aerosol and cloud property statistics to conduct the linear regression strongly reduces the variability in input parameters compared with a method using instantaneous daily satellite data, and thus underestimates the slopes of $\ln Nd$ versus $\ln AOD$ (Quaas et al., 2011).
- for the linear regression over a large region with meteorological fields varying significantly, the slope derived from monthly mean statistics tends to suffer more from the spatial covariation of aerosol and meteorological fields compared with instantaneous satellite observations.
- substantially reduced samples for monthly-based slopes would hamper the regression analysis.

Therefore, in order to avoid data missing as well as maintain a reliable regression analysis on a regional scale, adopting daily re-analysis is a more optimized method than using monthly average satellite observations.

As written already in the response above (comment on **Line 83**), the analysis for overcast clouds has also been conducted on a daily scale in this study. The result shows that there is no systematic difference in aerosol-cloud correlations between the analyses based on all cloud samples and 'overcast' clouds only.

Line 207 I agree that using monthly associations and not so much different from daily associations. And with monthly data there is added flexibility (e.g. the focus can be on the use of much higher quality data).

As stated in the main text, the sampling biases can affect derived RF_{aci} values through two pathways. First, it changes the regressions between Nd and AOD, which is the concern of the reviewer. The second pathway is by altering cloud fraction and cloud optical depth directly, key parameters in the calculation of RF_{aci} , as pure satellite-based investigations require the coexistence of retrievals of cloud and aerosol when computing RF_{aci} value (equation (3) in *Methods*).

Our discussion near line 207 in the initial manuscript is intended to demonstrate that, using monthly instead of daily τ_{ant} in the pure satellite-based analysis (Quaas et al., 2008; Ma et al., 2014; Patel et al., 2017) could be helpful to avoid the impact raised by the second pathway, which contributes the most to the increase in RF_{aci} .

Line 211/227 Yes, a cloud regime based analysis is a good next step, best if you can demonstrate that such a stratification does not matter.

Unfortunately, it is difficult at this moment to classify cloud types accurately with the passive remote sensing data used in this study. Due to strict revision time, we will investigate this in a future study by combining the active remote sensing. Also, cloud regime based analysis is beyond the scope of the current study, so it is just discussed as a potential next step.

Line 250+ I am confused. I would have expected this chapter earlier, to better understand the results of the previous chapters. I strongly encourage to move chapter 4 after the introduction.

According to NCOMMS format, the *Methods* section has to be placed last. But we have introduced the data and method briefly at the end of the introduction and referred to the *Methods* section for more details.

Line 280 just a simple fit... why not full radiative transfer? Does this work?

The original intention of this study is to estimate RF_{aci} relying on satellite observations alone, thus we do not use a radiative transfer model. Actually, the performance of multilinear regression fitting used here has been verified by previous studies. Their results showed that the fitted planetary albedo overall agrees very well with both CERES-retrieved albedo (Quaas et al., 2008) and the albedo simulated by a radiative transfer model (Patel et al., 2017). The associated discussion has been added to the *Methods* section (line 346-349).

Line 292 I prefer to define anthropogenic AOD via the fine-mode AOD fraction. Fine-mode AOD is available over oceans from the MODIS std product and from P.Ginoux over land. MISR provides fine-mode AOD over both land and oceans.

We thank the reviewer for this suggestion. The anthropogenic fraction has also been defined via the fine-mode AOD fraction ($f_{\text{ant-fine}}$; line 361-363) for computing RF_{aci} . Here, we choose to use the POLDER-3/GRASP fine-mode AOD product over both land and oceans, which has more observations than MISR and also has been reported to have good quantitative skill even over land (Li et al., 2020; Wei et al., 2020).

References

- Bai, H., Wang, M., Zhang, Z., & Liu, Y. (2020). Synergetic Satellite Trend Analysis of Aerosol and Warm Cloud Properties over Ocean and Its Implication for Aerosol- Cloud Interactions. *Journal of Geophysical Research: Atmospheres*, 125 (6), e2019JD031598.
- Bellouin, N., Quaas, J., Morcrette, J. J., & Boucher, O. (2013). Estimates of aerosol radiative forcing from the MACC re-analysis.

- Bréon, F. M., Tanré, D., & Generoso, S. (2002). Aerosol effect on cloud droplet size monitored from satellite. *Science* , 295 (5556), 834-838.
- Grandey, B. S., & Stier, P. (2010). A critical look at spatial scale choices in satellite-based aerosol indirect effect studies. *Atmospheric Chemistry and Physics Discussions* , 10 (6).
- Gryspeerdt, E., & Stier, P. (2012). Regime- based analysis of aerosol- cloud interactions. *Geophysical Research Letters*, 39, L21802. <https://doi.org/10.1029/2012GL053221>
- Grosvenor, D. P., Sourdeval, O., Zuidema, P., Ackerman, A., Alexandrov, M. D., Bennartz, R., ... & Deneke, H. (2018). Remote sensing of droplet number concentration in warm clouds: A review of the current state of knowledge and perspectives. *Reviews of Geophysics* , 56 (2), 409-453.
- Jia, H., Ma, X., Quaas, J., Yin, Y., & Qiu, T. (2019). Is positive correlation between cloud droplet effective radius and aerosol optical depth over land due to retrieval artifacts or real physical processes?. *Atmospheric Chemistry & Physics* , 19 (13).
- Jia, H., Ma, X., Yu, F., Liu, Y., & Yin, Y. (2019). Distinct impacts of increased aerosols on cloud droplet number concentration of stratus/stratocumulus and cumulus. *Geophysical Research Letters*, 46. <https://doi.org/10.1029/2019GL085081>
- Li, L., Che, H., Derimian, Y., Dubovik, O., Luan, Q., Li, Q., ... & An, L. (2020). Climatology of fine and coarse mode aerosol optical thickness over East and South Asia derived from POLDER/PARASOL satellite. *Journal of Geophysical Research: Atmospheres* , e2020JD032665.
- Ma, X., Yu, F., & Quaas, J. (2014). Reassessment of satellite- based estimate of aerosol climate forcing. *Journal of Geophysical Research: Atmospheres* , 119 (17), 10-394.
- O'Dell, C. W., Wentz, F. J., & Bennartz, R. (2008). Cloud liquid water path from satellite-based passive microwave observations: A new climatology over the global oceans. *Journal of Climate* , 21 (8), 1721-1739.
- Patel, P. N., Quaas, J., & Kumar, R. (2017). A new statistical approach to improve the satellite-based estimation of the radiative forcing by aerosol-cloud interactions. *Atmospheric Chemistry & Physics* , 17 (5).
- Quaas, J., Boucher, O., Bellouin, N., & Kinne, S. (2008). Satellite- based estimate of the direct and indirect aerosol climate forcing. *Journal of Geophysical Research: Atmospheres* , 113 (D5).
- Quaas, J., Boucher, O., Bellouin, N., & Kinne, S. (2011). Which of satellite-or model-based estimates is closer to reality for aerosol indirect forcing?. *Proceedings of the National Academy of Sciences* , 108 (46), E1099-E1099.
- Quaas, J., Arola, A., Cairns, B., Christensen, M., Deneke, H., Ekman, A. M. L., Feingold, G., Fridlind, A., Gryspeerdt, E., Hasekamp, O., Li, Z., Lipponen, A., Ma, P.-L., Mülmenstädt, J., Nenes, A., Penner, J., Rosenfeld, D., Schrödner, R., Sinclair, K., Sourdeval, O., Stier, P., Tesche, M., van Diedenhoven, B., and Wendisch, M.:

- Constraining the Twomey effect from satellite observations: Issues and perspectives, *Atmos. Chem. Phys. Discuss.*, <https://doi.org/10.5194/acp-2020-279>, in press, 2020.
- Randles, C. A., Da Silva, A. M., Buchard, V., Colarco, P. R., Darmenov, A., Govindaraju, R., ... & Shinozuka, Y. (2017). The MERRA-2 aerosol reanalysis, 1980 onward. Part I: System description and data assimilation evaluation. *Journal of Climate* , 30 (17), 6823-6850.
- Unglaub, C., Block, K., Mülmenstädt, J., Sourdeval, O., & Quaas, J. (2020). A new classification of satellite-derived liquid water cloud regimes at cloud scale. *Atmospheric Chemistry and Physics (Online)* , 20 (4).
- Várnai, T., & Marshak, A. (2009). MODIS observations of enhanced clear sky reflectance near clouds. *Geophysical Research Letters* , 36 (6).
- Vogelmann, A. M., McFarquhar, G. M., Ogren, J. A., Turner, D. D., Comstock, J. M., Feingold, G., et al. (2012). RACORO extended-term aircraft observations of boundary layer clouds. *Bulletin of the American Meteorological Society*, 93(6), 861–878.
- Wei, Y., Li, Z., Zhang, Y., Chen, C., Dubovik, O., Zhang, Y., ... & Ge, B. (2020). Validation of POLDER GRASP aerosol optical retrieval over China using SONET observations. *Journal of Quantitative Spectroscopy and Radiative Transfer* , 106931.
- Yu, F., Ma, X., & Luo, G. (2013). Anthropogenic contribution to cloud condensation nuclei and the first aerosol indirect climate effect. *Environmental Research Letters*, 8(2), 024029.
- Zhang, J., Reid, J., and Holben, B.: An analysis of potential cloud artifacts in MODIS over ocean aerosol optical thickness products, *Geophys. Res. Lett.*, 32, L15803, <https://doi.org/10.1029/2005GL023254>, 2005.

Reviewer #2 (Remarks to the Author):

Review of "Significant underestimation of radiative forcing by aerosol–cloud interactions derived from satellite-based methods" by H. Jia et al.

The authors present a compelling case for missing aerosol forcing in satellite based estimates due to sampling biases introduced when seeking to match cloud properties with (clear-sky) aerosol estimates. The authors decompose this bias in to a component due to a mis-estimation of the regression slope, and a component due to a mis-sampling of the underlying cloud property distributions. They seek to account for these biases using an assimilated aerosol product which is available under all-sky conditions. They examine potential deficiencies in this assimilated product and provide an updated satellite based estimate of R_{Faci}.

The paper addresses an important issue and is generally well structured. I am happy to recommend publication once the following points have been satisfactorily addressed.

Thanks to the reviewer for very helpful comments and suggestions, which have allowed us to clarify and improve the manuscript. We have revised the manuscript carefully according to the reviewer's comments. Please see the following detailed point-by-point responses.

Major concerns:

- Using an assimilated aerosol product provides a clear demonstration of the potential magnitude of the affect of the sampling biases discussed here, but the exact forcing values will depend on the details of the model, and in particular on the co-variability of aerosol and precipitation. This was most clearly demonstrated in Gryspeerdt et al. 2015 (<https://doi.org/10.5194/acp-15-7557-2015>), who showed the difficulty in representing the correlation between AOD and cloud properties in both observations and global models. How does MERRA-2 represent activation and wet-removal, how has this been validated and what affect could it have on the results presented here?

Aerosols in MERRA-2 are simulated with a radiatively coupled version of the Goddard Chemistry, Aerosol, Radiation, and Transport model (GOCART; Chin et al. 2002; Colarco et al. 2010). Wet scavenging processes in the model include rainout (in-cloud precipitation) and washout (below cloud precipitation) in large-scale precipitation and in deep convective cloud updraft. The in-cloud scavenging is parameterized directly by precipitation rate rather than cloud nucleation, thus aerosol activation treatments are not incorporated. Numerous studies have demonstrated the skill of the GOCART aerosol module in simulating AOD and other observable aerosol properties (e.g., Colarco et al. 2010; Nowottnick et al. 2010, 2011; Bian et al. 2013). It is acknowledged that the co-variability of aerosol and precipitation as well as the uncertainty of wet scavenging processes in the model could confound the interpretation of aerosol-cloud interactions. However, this study has been restricted to low liquid clouds, with liquid water path overall lower than 100 gm⁻² (Figure S8), thus generating very little rain. O'Dell et al. (2008) also found that non-precipitating

cloud pixels account for roughly 90% of all footprints globally. Therefore, the co-variability of aerosol and precipitation as demonstrated by Gryspeerd et al. (2015) and the treatment of wet removal are unlikely to significantly affect the results presented here. The associated discussions have been included in the revised manuscript (line 270-275).

- Further, while the authors have demonstrated that the MERRA-2 aerosol product does a reasonable job of reproducing MODIS AOD, it would be valuable to present (or cite a) validation against an independent instrument, especially given the importance of this to the manuscript.

Thanks for the suggestion. The performance of MERRA-2 AOD has been comprehensively validated by comparing with independent observations from ground, aircraft, and shipborne instruments (Randles et al., 2017; Gueymard and Yang, 2020). The relevant study has been cited in the revised manuscript (line 320-321).

- Also, the manuscript can be difficult to read at times and would benefit from editing help from someone with full professional proficiency in English.

We have tried to further improve the English language, but also Nature Communications will copy-edit the manuscript after acceptance.

Minor issues:

L25 - '...with more droplets.' This is only true for a fixed amount of liquid water.

Corrected.

L67 - Suggest replacing 'data-tied' with 'consistent'

Corrected.

L108 - 'have' -> has

Corrected.

L108 - Remove 'pretty' colloquialism

Corrected.

L110 - 'ones' -> 'clouds'

Corrected.

L207-208 - "Moreover, although... " This sentence compares RF_{aci} to RF_{aci} and doesn't seem to make sense.

Thanks for the reminder. The sentence has been rephrased as follows:

"Moreover, although the corrected slopes of $\ln N_d$ versus $\ln AOD$ only explain the 13% increase of annual averaged RF_{aci} , it can affect almost all satellite-related studies."

References

- Bian, H., Colarco, P. R., Chin, M., Chen, G., Rodriguez, J. M., Liang, Q., ... & Diskin, G. (2013). Source attributions of pollution to the Western Arctic during the NASA ARCTAS field campaign. *Atmospheric Chemistry and Physics* , 13 (9), 4707-4721.
- Bréon, F. M., Tanré, D., & Generoso, S. (2002). Aerosol effect on cloud droplet size monitored from satellite. *Science* , 295 (5556), 834-838.
- Chin, M., Rood, R. B., Lin, S. J., Müller, J. F., & Thompson, A. M. (2000). Atmospheric sulfur cycle simulated in the global model GOCART: Model description and global properties. *Journal of Geophysical Research: Atmospheres* , 105 (D20), 24671-24687.
- Colarco, P., da Silva, A., Chin, M., & Diehl, T. (2010). Online simulations of global aerosol distributions in the NASA GEOS- 4 model and comparisons to satellite and ground-based aerosol optical depth. *Journal of Geophysical Research: Atmospheres* , 115 (D14).
- Gryspeerd, E., Stier, P., White, B. A., & Kipling, Z. (2015). Wet scavenging limits the detection of aerosol effects on precipitation. *Atmospheric Chemistry & Physics* , 15 (13).
- Gueymard, C. A., & Yang, D. (2020). Worldwide validation of CAMS and MERRA-2 reanalysis aerosol optical depth products using 15 years of AERONET observations. *Atmospheric Environment*, 225, 117216.
- Nowottnick, E., Colarco, P., Ferrare, R., Chen, G., Ismail, S., Anderson, B., & Browell, E. (2010). Online simulations of mineral dust aerosol distributions: Comparisons to NAMMA observations and sensitivity to dust emission parameterization. *Journal of Geophysical Research: Atmospheres* , 115 (D3).
- Nowottnick, E., Colarco, P., Silva, A. D., Hlavka, D., & McGill, M. (2011). The fate of saharan dust across the atlantic and implications for a central american dust barrier. *Atmospheric Chemistry and Physics* , 11 (16), 8415-8431.
- O'Dell, C. W., Wentz, F. J., & Bennartz, R. (2008). Cloud liquid water path from satellite-based passive microwave observations: A new climatology over the global oceans. *Journal of Climate* , 21 (8), 1721-1739.
- Randles, C. A., Da Silva, A. M., Buchard, V., Colarco, P. R., Darmenov, A., Govindaraju, R., ... & Shinozuka, Y. (2017). The MERRA-2 aerosol reanalysis, 1980 onward. Part I: System description and data assimilation evaluation. *Journal of Climate* , 30 (17), 6823-6850.

Reviewer #3 (Remarks to the Author):

The authors put forward a straight-forward, but impactful estimate of the RF_{aci} . By neglecting clouds with high coverage and optical depth we underestimate how strong RF_{aci} is. The authors correct this and move toward reconciling literature estimates of RF_{aci} . While aerosol column brightness based estimates have many well known problems, it is important to characterize why older estimates of RF_{aci} tend to be weak compared to newer estimates and GCMs.

We thank the reviewer for taking the time to assess the manuscript and for providing helpful comments and suggestions to improve the manuscript. We have revised the manuscript carefully according to the reviewer's comments. Please see the following detailed point-by-point responses.

While I feel this study may merit publication in NCOMMS I think the addition of a GCM-only analysis to support their observational analysis following Ma et al. [2018] is needed to make sure it is robust. Their analysis can be replicated by sampling data from a GCM running a satellite simulator easily.

Satellite simulator is developed for facilitating model-to-satellite comparisons in an apples-to-apples way by converting model outputs to satellite-like diagnostics. The use of satellite simulator by Ma et al. (2018) was intended to determine the individual contribution of different satellite retrieval limitations to the discrepancy of cloud susceptibilities between satellite and model. This was achieved by incrementally accounting for different components of retrieval algorithms (such as detection threshold of aerosol layer, aerosol typing algorithm, and cloud retrieval assumptions) in a satellite simulator. The relevant findings by Ma et al. (2018) have been discussed in detail in the revised manuscript (line 43-45). The goal of our study is to quantify the effect of sampling biases of satellite observations on the estimate of RF_{aci} , with a focus on the comparison of the conventional and improved satellite-based methods, not involving the direct comparison with model as well as any changes in retrieval algorithms as Ma et al. (2018) did. Therefore, adopting re-analysis product offering the full coverage of aerosol distribution would be sufficient for this kind of analysis, and our observation-based analyses have also clearly demonstrated the effect of sampling biases, though the exact forcing values are also affected by many other uncertainties as detailed in the discussion section. From the perspective of GCM approach, it would of course be a necessity to accurately estimate RF_{aci} by employing the GCM simulations using satellite simulator. In future, if possible, we will be grateful to work on this issue with the GCM groups using satellite simulator, e.g. Ma et al. (2018).

The authors neglect a lot of the existing literature on how to appropriately retrieve N_d [Daniel P. Grosvenor et al., 2018] and utilize an unusual retrieval methodology (using CERES observations). Their decision to use this data set instead of a more well-established one must be discussed.

Thanks for your suggestions. The discussion on how to appropriately retrieve Nd has been inserted into the *Methods* section, and the relevant references have been cited as well (line 340-342).

In the current study, we utilized CERES observations for two reasons:

- The CERES product provides strictly collocated cloud properties and TOA radiative fluxes so that we can obtain more reliable fitting parameters of the multilinear regression that links radiation with cloud properties, and thus a more accurate estimate of RF_{aci} . For pixel-level MODIS data, however, there is a need to interpolate CERES radiative observations to MODIS resolution (1x1 km²), which would introduce additional uncertainties.
- Also, previous satellite-based studies on the subject have made use of CERES cloud properties to estimate RF_{aci} (Quaas et al., 2008; Bellouin et al., 2013; Ma et al., 2014). Therefore, using CERES data here helps to compare with previous results.

We acknowledge that using well-validated MODIS pixel-level Nd can reduce the uncertainties in aerosol-cloud correlations introduced by cloud retrieval biases. In addition, the large amounts of data from MODIS pixel-level retrievals allow one to conduct statistical analysis on a smaller spatial scale, thus greatly avoiding the co-variations of aerosol and meteorological conditions over large regions. Given all that, adopting MODIS Nd would be a nice attempt in future works. These discussions have also been included in the revised manuscript (line 263-265; line 293-295). However, the global long-term MODIS pixel-level data takes a lot of time to download and analyze. Due to strict revision time, we will conduct MODIS-based analysis in a future study.

Major:

P2 L29: I think the distinction needs to be made that these are estimates based on covariability between column measurements of aerosol and cloud properties. Estimates based on metrics like aerosol mass tend to have higher values [Bellouin et al., 2020; Boucher and Lohmann, 1995; McCoy et al., 2018].

Thanks for the suggestion. We have rephrased the sentence in the revised manuscript as follows:

“The satellite-based RF_{aci} using column measurements of aerosol and cloud properties, typically in the range of -0.2 to -0.7 $W m^{-2}$, is much weaker than modeled values of -0.3 to -1.8 $W m^{-2}$. Though observational estimates based on metrics like aerosol mass tend to generate higher RF_{aci} (-0.97 ± 0.23 $W m^{-2}$), they are still generally lower than modeled values.”

P2 L36: Authors need to cite and discuss Ma et al. [2018], which shares a lot of the same techniques and goals as this paper but focusses on aerosol-cloud adjustments.

We thank the reviewer for this suggestion. The study by Ma et al. [2018] has been cited here, and relevant findings have been discussed. (line 43-45)

P2 Ln 60: the argument is unclear here and appears to conflate adjustments and RF_{aci} . It seems like all the authors are saying is that $d\ln Nd/d\ln AOD = f_n(CF, \tau_{cloud})$. Is the second argument that clouds with a larger cf or lower τ would tend to have a larger radiative response to changes in Nd ? This is accounted for in many studies estimating RF_{aci} .

Thanks for the reminder. Sorry for the inaccurate expression. Here, we intended to say that, the sampling biases systematically exclude clouds with high cloud fraction and optical depth, which exert a stronger cooling effect by reflecting more incoming shortwave radiation than thin and broken cumulus clouds. Missing these clouds in the calculation of RF_{aci} (see equation (3) in *Methods*), will lead to a lower RF_{aci} .

The sentence has been rephrased as follows:

*“The second pathway is by altering f and cloud optical depth (τ_c) directly, key parameters in the calculation of RF_{aci} (see equation (3) in *Methods*)”*

L36 the authors here and elsewhere suggest that higher τ clouds would have a larger RF_{aci} . The opposite should be true. When clouds have a low τ you get more bang for your buck for a set increase in τ (see fig. 3 of Carslaw et al. [2013]). It might be useful to show the analysis Fig. S6 into a changed τ and changed CF contribution.

The reviewer (also in Carslaw et al. [2013]) means the increase of cloud optical depth (τ_c) with aerosol loading becomes slow for high τ_c conditions, where lower sensitivity of τ_c to aerosols occurs, which is relevant to the nonlinear aerosol-cloud relationship. Here, however, we mean that clouds with high cloud fraction and τ_c can reflect more incoming shortwave radiation than thin and broken clouds, thus exert a stronger cooling effect, in which cloud fraction (f) and τ_c are directly linked to RF_{aci} (see equation (3) in *Methods*).

According to the reviewer's suggestion, we further calculated the individual contributions of changed cloud fraction and τ_c to RF_{aci} . With *Aero_Cld* as a control test, we increased cloud fraction (*Aero_Cld_f*) and τ_c (*Aero_Cld_τc*), respectively, according to their relative changes from *Aero_Cld* to *All_Cld* shown in Figure 1. Figure R5 shows the annual mean RF_{aci} for *Aero_Cld*, *Aero_Cld_f* and *Aero_Cld_τc*, respectively. It is illustrated that the RF_{aci} values generally increase with both cloud fraction and τ_c , except for the tropical ocean where f tends to decrease after including all cloud samples (Figure 1). It should be noted that quantifying the individual changes of RF_{aci} by simply scaling f and τ_c by the overall relative changes, would not offer accurate contributions due to the nonlinearity between RF_{aci} and f & τ_c (equation (3) in *Methods*). The analysis here just provides a sense of the direction of the individual changes of RF_{aci} , so it is not added to Figure S6.

Figure R5. Annual mean first indirect forcing (RF_{aci}) for *Aero_Cld* (a), *Aero_Cld_f* (b) and *Aero_Cld_tc* (c), respectively.

Ln73: Why use CERES for cloud properties? This is highly unusual and there has no validation of the Nd from CERES. MODIS Nd could be used, which would match the MODIS AOD being used, and which has been extensively validated and discussed elsewhere [Daniel P. Grosvenor et al., 2018]. On Ln 251 the CERES data being used is discussed, but it is unclear what data is removed, such as high SZA, or low CF [D. P. Grosvenor and Wood, 2014]. This has a substantial impact on the study and needs to be clearly stated.

For the reasons using CERES rather than MODIS cloud properties, please refer to the response above. Though not extensively as MODIS, the performance of CERES cloud retrievals has been validated by comparing with independent ground-based measurements (Dong et al., 2008; Minnis et al., 2011; Xi et al., 2014).

To obtain more reliable fitting parameters, we excluded the retrievals involving bright surfaces, high solar zenith angle ($>65^\circ$), thin clouds (liquid water path $< 20 \text{ g m}^{-2}$), and multilayered clouds, which has been clarified in the revised manuscript (line 355-357). As for cloud fraction (f) screening, we made an initial attempt to ensure 'overcast' clouds by restricting retrievals to clouds with $f > 80 \%$. This threshold was also adopted by previous studies (Bai et al., 2018; Grosvenor et al., 2018). It is found that there is no systematic difference in aerosol-cloud correlations between the analyses based on all cloud samples and 'overcast' clouds only (Figure R4). Thus, we did not further limit cloud fraction for the main results.

Ln 109 cite Hamilton et al. [2014]

Thanks. The reference has been cited.

Ln 221 I believe the authors are conflating systematic bias and random uncertainty in their citation and discussion. Random uncertainty in Nd was reported at 80% on a pixel level in Daniel P. Grosvenor et al. [2018] and this dropped substantially when averaged to a 1x1° region. However, as stated in that review paper the systematic bias in the Nd relative to in situ was low. The random uncertainty discussed here does not affect the results of the present publication beyond increasing uncertainty in their regression coefficients.

Thanks for the reminder. In the revised manuscript, we no longer associate aerosol-cloud correlations with the random uncertainty bias in Nd.

Minor:

P1 L12: I think that the authors mean the uncertainty due to differing estimates of RFaci from GCMs and obs.

Thanks for the reminder. We have rephrased the sentence in the revised manuscript as follows:

“Satellite-based estimates of radiative forcing by aerosol–cloud interactions (RFaci) are consistently smaller than those from global models, hampering accurate projections of future climate change.”

P1 L17: Increases in magnitude or becomes more positive?

Thanks for the reminder. It means the increase in magnitude. Have clarified.

P2 L28 ‘in’

Corrected.

Bellouin, N., et al. (2020), Bounding Global Aerosol Radiative Forcing of Climate Change, Rev. Geophys., 58(1), doi: 10.1029/2019rg000660.

Boucher, O., and U. Lohmann (1995), The sulfate-CCN-cloud albedo effect, Tellus B, 47(3), 281-300, doi: 10.1034/j.1600-0889.47.issue3.1.x.

Carslaw, K. S., et al. (2013), Large contribution of natural aerosols to uncertainty in indirect forcing, Nature, 503(7474), 67-71, doi: 10.1038/nature12674.

Grosvenor, D. P., and R. Wood (2014), The effect of solar zenith angle on MODIS cloud optical and microphysical retrievals within marine liquid water clouds, Atmos. Chem. Phys., 14(14), 7291-7321, doi: 10.5194/acp-14-7291-2014.

Grosvenor, D. P., et al. (2018), Remote Sensing of Droplet Number Concentration in Warm Clouds: A Review of the Current State of Knowledge and Perspectives, Rev. Geophys., 56(2), 409-453, doi: 10.1029/2017rg000593.

Hamilton, D. S., L. A. Lee, K. J. Pringle, C. L. Reddington, D. V. Spracklen, and K. S. Carslaw (2014), Occurrence of pristine aerosol environments on a polluted planet, *Proceedings of the National Academy of Sciences*, 111(52), 18466-18471, doi: 10.1073/pnas.1415440111.

Ma, P.-L., P. J. Rasch, H. Chepfer, D. M. Winker, and S. J. Ghan (2018), Observational constraint on cloud susceptibility weakened by aerosol retrieval limitations, *Nat. Commun.*, 9(1), 2640, doi: 10.1038/s41467-018-05028-4.

McCoy, D. T., F. A. M. Bender, D. P. Grosvenor, J. K. Mohrmann, D. L. Hartmann, R. Wood, and P. R. Field (2018), Predicting decadal trends in cloud droplet number concentration using reanalysis and satellite data, *Atmos. Chem. Phys.*, 18(3), 2035-2047, doi: 10.5194/acp-18-2035-2018.

References

Bai, H., Wang, M., Zhang, Z., & Liu, Y. (2020). Synergetic Satellite Trend Analysis of Aerosol and Warm Cloud Properties over Ocean and Its Implication for Aerosol- Cloud Interactions. *Journal of Geophysical Research: Atmospheres* , 125 (6), e2019JD031598.

Bellouin, N., Quaas, J., Morcrette, J. J., & Boucher, O. (2013). Estimates of aerosol radiative forcing from the MACC re-analysis.

Dong, X., Minnis, P., Xi, B., Sun- Mack, S., & Chen, Y. (2008). Comparison of CERES- MODIS stratus cloud properties with ground- based measurements at the DOE ARM Southern Great Plains site. *Journal of Geophysical Research: Atmospheres* , 113 (D3).

Grosvenor, D. P. et al. (2018). Remote sensing of droplet number concentration in warm clouds: A review of the current state of knowledge and perspectives. *Reviews of Geophysics* , 56 (2), 409-453.

Ma, X., Yu, F., & Quaas, J. (2014). Reassessment of satellite- based estimate of aerosol climate forcing. *Journal of Geophysical Research: Atmospheres* , 119 (17), 10-394.

Minnis, P. et al. (2011). CERES edition-2 cloud property retrievals using TRMM VIRS and Terra and Aqua MODIS data—Part II: Examples of average results and comparisons with other data. *IEEE transactions on geoscience and remote sensing* , 49 (11), 4401-4430.

Quaas, J., Boucher, O., Bellouin, N., & Kinne, S. (2008). Satellite- based estimate of the direct and indirect aerosol climate forcing. *Journal of Geophysical Research: Atmospheres* , 113 (D5).

Xi, B., Dong, X., Minnis, P., & Sun- Mack, S. (2014). Comparison of marine boundary layer cloud properties from CERES- MODIS Edition 4 and DOE ARM AMF measurements at the Azores. *Journal of Geophysical Research: Atmospheres* , 119 (15), 9509-9529.

Review of revised paper:

Significant underestimation of radiative forcing by aerosol-cloud interactions derived from satellite based methods

by H. Jia et al.

General comments

The paper represents an interesting approach in data analysis and merits a publication.

This said, I still struggle with some of the statements and conclusions. There are still many uncertain elements involving the 'slopes', in particular assumptions to anthropogenic fractions to (AOD and) AODf and along with it assumptions to the pre-industrial reference. Also question to proper representation by (Merra) simulated (AOD and) AODf remain.

The papers central theme is that matching aerosol and cloud retrievals in 100x100km regions have to ignore many cloudy regions by using daily/instantaneous, which eventually – according to the authors - translate in underestimates for aerosol impacts on clouds (ACI impacts). They argue that with larger cloud optical depths (COT) in the omitted regions, aerosol associated planetary albedo increases are higher. But, this is not that obvious to me. For instance a 20% increase in COT yields the same planetary albedo increase at lower COT (5→6) as at higher COT (15→18).

Another way to sidestep this bias (to a large degree) would be the use of monthly aerosol/cloud associations. In the case of monthly averages also short term feedbacks or delayed responses would be included and given that 20 years of monthly data now exist the argument about poor statistics fades. The authors already offered some comparisons between monthly and daily statistics, it would be great (to argue their claim) if they could subsample monthly statistics, to only 100*100km regions where daily associations were possible and then compare those monthly 'slopes' (A) to 'slopes' (B) involving all monthly associations. If the slopes of (B) are significantly larger than (A) they more likely have a case.

On a more general note, I wonder why $\ln(Nd/\ln(AODf))$ slopes and not $\ln(Nd)/AODf$ slopes are used, because the approach with $\ln(AODf)$ gives a lot of weight to small AODf changes, which are much more uncertain in absolute sense than larger AODf changes. Hereby it should be considered that for very low AOD and AODf retrievals are very noisy and usually have a tendency to overestimate AOD (e.g. MODIS), whereas modeling generally lacks long-range transport, yielding often AOD underestimates in remote (low AOD) regions.

I really appreciate, that the authors responded to all my concerns with many interesting answers. So let me re-respond:

The authors demonstrate that for regional averages cloud retrieval quality filters have only a small impact, which is a bit surprising to me. Still in Figure S7 they show a in most regions a tendency to steeper slopes without quality filters. I found that the cloud (e.g. CDNC) data scatter without quality filters is much larger.

I agree that regions with high cloud fractions, often will not allow aerosol retrievals, even on a 100km scale for daily data. So indeed cases with larger cloud fields would not be considered. But it is not so clear to me that there should be automatically a slope underestimate with higher COT.

One bias argument here is that “(missed) stratiform clouds are more sensitive than cumulus clouds”. I am not so convinced in terms of the first indirect effects but if the lifetime effect is included, then certainly I would agree. By the way, for many stratiform clouds also the (often lacking) co-location in vertical space with aerosol (e.g. wildfire outflow over oceans) matters.

The other argument here is that “(water) clouds with high fractions and COT exert a stronger cooling” (e.g. a higher solar albedo). I am not convinced that this is true, as the susceptibility is maxed at COT~10, and COTs of 5 and 15 have a similar (and smaller) solar reflection response for the same %age COT change.

Certainly the cloud-fraction in the (missed) stratiform clouds is higher, but would not this also be considered, when applying valid $d\ln(N_d)/d\ln(AOD_f)$ relationships to stratiform clouds?

I do not quite follow the “biased high retrieved AOD within cloudy pixels” argument, unless the $d\ln(N_d)/d\ln(AOD_f)$ slope is meant. If an AOD (or better AOD_f) increase is higher, then it yields more CDNC and a higher ACI. But why would someone use biased AOD data anyway?

The relatively large response over land (compared to near-by oceans) is argued that the “anthropogenic AOD fraction is much larger over land” and nicely documented in Figure 4. I really appreciate this plot, because it is so important for the derived ACI impact, because a high anthrop. fraction determines not only the anthropogenic increase but also the pre-industrial background. The now applied anthropogenic AOD and AOD_f fractions, however, are very high in NH industrial regions, compared to what AeroCom modeling suggests (see figure below), especially if a 1850 reference year is applied.

While anthropogenic fraction is quite revealing, it does not tell the entire story, as it is only a relative property, whereas the product (with AOD and AOD_f) matters. Thus, a map showing the pre-industrial AOD or AOD_f and the anthropogenic AOD or AOD_f would very insightful (if not in the main text, so at least in the supplement). In fact the anthropogenic fraction difference in Figure 4 and the is not so interesting.

I agree that the impact from the solar insolation is relatively minor (although it will certainly matter for the winter season over Europe).

I really like that the authors now also work with the AOD_f (which is a better proxy for aerosol number than AOD). By the way the authors also applied AI (should be similar in response to AOD_f, except that Angstrom values are usually terrible especially from retrievals) and sulfate mass (sulfate is the most likely droplet nuclei) in the Appendix and demonstrated differences to slopes using just AOD – and these differences should largely mimic AOD_f vs AOD based slope differences. With respect to comparing AOD_f and AOD, the much lower slope in Figure 2a over the northern hemispheric industrial regions by POLDER data makes me wonder, because for ‘cld’ with Merra in Figure 3c these are the regions with the largest ACI contributions. Also by replacing AOD with AOD_f in the analysis the estimated ACI impact for total AOD in Figure 3d (-0.67 global) almost doubles (-1.22 global) in Figure 4d, mainly, as I see it, with significant larger contributions over NH oceans, although there are now larger contributions over continents as well. Since ACI impacts increased everywhere, I wonder if this is possibly related to a cleaner pre-industrial background, by applying pre-industrial AOD_f rather than pre-industrial AOD?

With respect to differences in slopes between land and ocean, I think that aerosol number is the most important element and other properties (e.g. chem composition) at least for the fine-mode are only secondary. Interestingly, the regional slope comparisons in Figure 2 show a very strong inter-land and inter-ocean region variability, whereas on average all slopes over land and ocean are not so much different.

Thanks for comparing daily slopes with monthly slopes in Figure R2. I would interpret the results, that the monthly slopes are usually larger over the important northern hemispheric land and

ocean regions for the Pacific and Asia (as I had expected) except for the more northern located Atlantic and Europe, where missing winter data may contribute (although during high latitude winters - when there is little to no sun - ACI contributions should be small).

The mentioned concerns on applying monthly statistics are noted. I still think though that such an analysis has some merit. Satellite retrievals now cover almost from 2 decades a potentially poor statistical aspect not be such a concern, especially if applied to larger regions. And while it is clear that more instantaneous data have more variability, local ACI impacts are often overestimates (e.g. shiptracks) and ACI responses are not always immediate often with delays of several days, so monthly averages possibly could better capture the 'effective' response.

Supplement:

The figure below in the left column presents anthropogenic AODf fractions from AeroCom modeling, generally larger fractions with an older (year 1750, AeroCom 1 emissions) reference and smaller fractions with a more recent (year 1850, AeroCom 2 emissions) reference. Using the 1850 reference (lower left) anthropogenic AODf contributions over the industrial areas are much lower than for the 1750 reference and much lower than those assumed by the authors in Figure 4b. A lower anthropogenic AODf fraction also means a much stronger pre-industrial background, and both elements reduce ACI impacts. In summary, without confidence in the anthropogenic AODf fraction large ACI uncertainties will remain.

MACv2: anthropogenic fraction of AODf today- AODf 1750 (AeroCom1 models, upper left) and anthropogenic fraction of AODf today- AODf 1850 (AeroCom2 models, lower left), anthropogenic BC fraction (upper right) and anthropogenic dust fraction (lower right)

Minor comments

160 using Polder is a good idea, but there are limited years (MISR also provides AODf globally for the entire Terra data record ... but is spatially sparse). Still in the analysis simulated MERRA AODf were preferred, which clouds the entire process. Have there been MERRA data comparisons at successful Polder retrievals? Do they statistically agree?

163 ... why 30% of the seasalt AOD in the MERRA AODf? Is there a good reason?

Reviewer comments, second round: –

Reviewer 1 - (previous page)

Reviewer #2 (Remarks to the Author):

I am satisfied that the authors have addressed all of my previous concerns and am happy to accept the paper in its current form.

Reviewer #3 (Remarks to the Author):

I thank the authors for their careful consideration of my comments and clarifications to their document. I am satisfied with their responses.

There seems to be a typesetting issue on page 5 with the entire page in heading font.

Thanks to the reviewers for taking the time to assess the manuscript and for providing thoughtful comments and suggestions to improve the manuscript. Below we address the reviewers' comments, with the reviewer comments in black, and our responses in blue. We have revised the manuscript accordingly.

REVIEWER COMMENTS

Reviewer #1 (Remarks to the Author):

Review of revised paper:

Significant underestimation of radiative forcing by aerosol-cloud interactions derived from satellite based methods

by H. Jia et al.

General comments

The paper represents an interesting approach in data analysis and merits a publication.

This said, I still struggle with some of the statements and conclusions. There are still many uncertain elements involving the 'slopes', in particular assumptions to anthropogenic fractions to (AOD and) AODf and along with it assumptions to the pre-industrial reference. Also question to proper representation by (Merra) simulated (AOD and) AODf remain.

We thank the reviewer for the suggestions. As suggested by the reviewer, monthly aerosol-cloud associations are now employed to compute the 'slope' and the corresponding RF_{aci} . Also, we adopt the anthropogenic fractions from AeroCom 1 and AeroCom 2 models as mentioned by the reviewer, and discussed the sensitivity of RF_{aci} to the choice of anthropogenic fraction in the revised manuscript. With respect to the concern of proper representation by MERRA-2 AODf, a comparison between POLDER-3 and MERRA-2 products is presented. Please see the following point-by-point responses for more details.

The papers central theme is that matching aerosol and cloud retrievals in 100x100km regions have to ignore many cloudy regions by using daily/instantaneous, which eventually – according to the authors - translate in underestimates for aerosol impacts on clouds (ACI impacts). They argue that with larger cloud optical depths (COT) in the omitted regions, aerosol associated planetary albedo increases are higher. But, this is not that obvious to me. For instance a 20% increase in COT yields the same planetary albedo increase at lower COT (5->6) as at higher COT (15->18).

We greatly appreciate the comment and explanation. We agree with the reviewer that at larger cloud optical depth (COT) the susceptibility for planetary albedo change is reduced, and thus the increase in COT is unlikely to contribute much to the increased RF_{aci} . In the revised manuscript, the enhanced RF_{aci} by correcting sampling biases is now attributed to the increased

cloud fraction only, since clouds with higher cloud fraction exert a stronger cooling effect, and are more sensitive to aerosol perturbations. Figure 1b has been removed from the revised manuscript and the text has been modified accordingly.

Another way to sidestep this bias (to a large degree) would be the use of monthly aerosol/cloud associations. In the case of monthly averages also short term feedbacks or delayed responses would be included and given that 20 years of monthly data now exist the argument about poor statistics fades. The authors already offered some comparisons between monthly and daily statistics, it would be great (to argue their claim) if they could subsample monthly statistics, to only 100*100km regions where daily associations were possible and then compare those monthly 'slopes' (A) to 'slopes' (B) involving all monthly associations. If the slopes of (B) are significantly larger than (A) they more likely have a case.

As suggested by the reviewer, we subsample monthly N_d and MERRA-2 AOD statistics from daily data according to four strategies as indicated by different subscripts (*AllCld*, *AeroCld*, *clear sky*) in Figure R1 (also Figure S8). It is clearly demonstrated that 'slopes' (B) (the slopes of $\ln N_{dAllCld}$ versus $\ln AOD_{AllCld}$) are overall larger than 'slopes' (A) (the slopes of $\ln N_{dAeroCld}$ versus $\ln AOD_{AeroCld}$), which is consistent with our daily-based conclusion.

We agree that the use of monthly aerosol/cloud statistics from satellite observations would be another potential way to sidestep the sampling biases. We have now in the revision also adopted monthly POLDER-3 AOD_f and monthly cloud properties to estimate RF_{aci} (Figure 4d). The results show that global-averaged RF_{aci} (-1.02 W m^{-2}) is close to MERRA-2 daily-based one (-1.09 W m^{-2} ; Figure 4c) but with different spatial distributions, i.e., much smaller RF_{aci} over land, which is also deviated from the modeled distribution (Figure 3e). A comparison of the slopes of $\ln N_d$ versus $\ln AOD_f$ between two cases shows that POLDER-3 monthly-based slopes are significantly lower than MERRA-2 daily-based ones over lands (Figure R2, also Figure S7). It should be noted that the use of monthly statistics can only avoid the sampling bias on clouds, and aerosol information associated with the clouds that fully cover $1^\circ \times 1^\circ$ grid box are impossible to be sampled. That is, this monthly AOD_f would not be well representative of the amount of aerosols actually linking with monthly N_d , especially over the regions where aerosols have large inter-daily variability. Figure R1 demonstrates that the slopes decrease accordingly as monthly AOD gradually become irrelevant to monthly $N_{dAllCld}$ (from AOD_{AllCld} to $AOD_{clear\ sky}$), even though all clouds have been sampled. The result here implies that the conclusion regarding spatial distribution (land-sea contrast) of RF_{aci} , should be drawn carefully if monthly statistics are applied. The associated discussions have been added in the revised manuscript (lines 239-251; lines 272-276).

Figure R1. Annual averaged slopes of the linear regression $\ln N_d$ versus $\ln AOD$ based on monthly aerosol-cloud associations for the regions over (a) land and (b) ocean. Here, monthly N_d and AOD statistics are subsampled from daily data according to four strategies as indicated by different subscripts (*AllCld*, *AeroCld*, *clear sky*). $N_{dAllCld}$ (AOD_{AllCld}) represents the monthly N_d (AOD) calculated from the data including all clouds. $N_{dAeroCld}$ ($AOD_{AeroCld}$) represents the monthly N_d (AOD) calculated from the subset of data where aerosol and cloud retrievals are available simultaneously. $AOD_{clear\ sky}$ is the monthly AOD for clear sky ($f = 0$ within $1^\circ \times 1^\circ$ grid box), which does not actually interact with clouds. The annual averaged slope is the average of the monthly slopes. The standard deviation of the inter-monthly variability of the regression slopes is shown as an error bar.

Figure R2. Annual averaged slopes of the linear regression $\ln N_d$ versus $\ln AOD_f$ based on POLDER-3 monthly (grey) and MERRA-2 daily (*AllCld* scenario; blue) statistics. The annual averaged slope here is the average of the monthly slopes. The standard deviation of the inter-monthly variability of the regression slopes is shown as an error bar.

On a more general note, I wonder why $\ln(N_d/\ln(AOD_f))$ slopes and not $\ln(N_d)/AOD_f$ slopes are used, because the approach with $\ln(AOD_f)$ gives a lot of weight to small AOD_f changes, which are much more uncertain in absolute sense than larger AOD_f changes. Hereby it should be considered that for very low AOD and AOD_f retrievals are very noisy and usually have a tendency to overestimate AOD (e.g. MODIS), whereas modeling generally lacks long-range transport, yielding often AOD underestimates in remote (low AOD) regions.

Thanks for the comment. $\ln(N_d)/\ln(AOD_f)$ slopes are used in order to investigate relative changes (i.e. changes in N_d with respect to $\ln AOD$ changes), to account for the logarithmic (saturating) behavior of the relation. But using $\ln(N_d)/AOD_f$ slopes can not do that. We agree

with the reviewer that the $\ln(N_d)/\ln(\text{AOD}_f)$ slope is more sensitive to small AOD_f changes, which is quite uncertain for both retrievals and modeling. To avoid this problem, we adopt the method by Hasekamp et al. (2019), leaving out the lowest 15% of data for AOD and AOD_f for the calculation of slopes in the revised manuscript. The discussions have been inserted in the revised manuscript (lines 404-406). The updated slopes and RF_{aci} are overall slightly decreased, but do not influence the conclusions in the manuscript. All associated statistics, tables and figures have been revised accordingly.

I really appreciate, that the authors responded to all my concerns with many interesting answers. So let me re-respond:

The authors demonstrate that for regional averages cloud retrieval quality filters have only a small impact, which is a bit surprising to me. Still in Figure S7 they show a in most regions a tendency to steeper slopes without quality filters. I found that the cloud (e.g. CDNC) data scatter without quality filters is much larger.

The little difference between slopes with and without cloud retrieval quality filters would be partly due to the relatively large spatial scale ($20\text{km} \times 20\text{km}$ region) where quality filters are applied. Within this large grid box, though restricting retrievals to clouds with fractions $> 80\%$, the biased retrievals on pixel-level grids ($1\text{km} \times 1\text{km}$) are still likely to be included. Therefore, with the use of CERES data in this study, which does not have any pixel-level flag identifying overcast or partly cloudy retrievals, it is not possible to perfectly conduct retrieval quality filters. As discussed in section 3 (lines 305-311), we believe that adopting pixel-level cloud retrievals with the overcast flag (such as MODIS level-2 $1 \times 1\text{km}^2$ product) in future investigations would be useful to demonstrate whether such filters matter a lot to aerosol-cloud correlations.

I agree that regions with high cloud fractions, often will not allow aerosol retrievals, even on a 100km scale for daily data. So indeed cases with larger cloud fields would not be considered. But it is not so clear to me that there should be automatically a slope underestimate with higher COT.

Thanks for the comment. The increased slope by including all missed clouds is due to the presence of more stratiform clouds, which have been reported to be more sensitive to aerosol perturbations (Gryspeerdt et al., 2012; Jia et al., 2019). Stratiform clouds are typically characterized by high cloud fraction, thus a larger slope would occur with a higher cloud fraction. We acknowledge that the larger slope would not be linked directly to higher COT, and the apparent connection is possibly modulated by the cloud fraction (those missed clouds have high cloud fraction and COT simultaneously). Therefore, in the revised manuscript, we only focus on the influence of cloud fraction, which is more straightforward with cloud types. The text has been revised accordingly.

One bias argument here is that "(missed) stratiform clouds are more sensitive than cumulus clouds". I am not so convinced in terms of the first indirect effects but if the lifetime effect is included, then certainly I would agree. By the way, for many stratiform clouds also the (often lacking) co-location in vertical space with aerosol (e.g. wildfire outflow over oceans) matters.

In addition to satellite-based study (Gryspeerdt et al., 2012), the more reliable in situ aircraft measurements (Vogelmann et al., 2012; Jia et al., 2019a) also observed the much stronger

increase in N_d with increased CCN for stratocumulus (Sc) than cumulus (Cu), i.e., the first indirect effects. Although higher CCN number concentration (N_{CCN}) favors the cloud droplet formation by supplying CCNs, higher N_{CCN} also lowers the maximum supersaturation at cloud base by generating more droplets to compete for the available humidity (Chen et al., 2016; Hudson & Noble, 2014a, 2014b). Therefore, how N_d responds to the increasing N_{CCN} is contingent on the competition between these two impacts. Our previous work revealed that the different first indirect effect between Sc and Cu was due to the different degree of reduction in cloud supersaturation caused by increasing aerosols (Jia et al., 2019a). More details can be found in Jia et al.(2019a), which has been cited in the main text.

We agree that lacking co-location in vertical space with aerosol matters to stratiform clouds over some specific regions. But this problem also remains for cumuliform clouds. It would be interesting to explore how the co-location in vertical affects the observed different first indirect effects between stratiform and cumuliform clouds in future investigations.

The other argument here is that “(water) clouds with high fractions and COT exert a stronger cooling” (e.g. a higher solar albedo). I am not convinced that this is true, as the susceptibility is maxed at COT~10, and COTs of 5 and 15 have a similar (and smaller) solar reflection response for the same %age COT change.

Thanks for the reminder. We agree with this. As responded above, the COT-associated arguments have been removed from the revised manuscript.

Certainly the cloud-fraction in the (missed) stratiform clouds is higher, but would not this also be considered, when applying valid $\ln(N_d)/\ln(AOD_f)$ relationships to stratiform clouds?

The reviewer is right. However, one aspect is not considered, which is that also the $\ln(N_d)/\ln(AOD_f)$ are affected by considering or not the stratiform clouds, and it is this aspect that is a key finding in the present study.

I do not quite follow the “biased high retrieved AOD within cloudy pixels” argument, unless the $\ln(N_d)/\ln(AOD_f)$ slope is meant. If an AOD (or better AOD_f) increase is higher, then it yields more CDNC and a higher ACI. But why would someone use biased AOD data anyway?

One of inherent satellite retrieval limitations is that, AOD could be overestimated due to either cloud contaminations where spurious clouds might be present in pixels that are erroneously identified as completely clear pixels (Zhang et al., 2005), or cloud adjacency effect where cloud-free pixels are brightened by reflected light from surrounding clouds (Várnai and Marshak, 2009). Meanwhile, cloud retrievals applied to partially cloudy and 3-D shaped clouds pixels are expected to deviate from the retrieval assumptions of overcast homogenous cloud and 1D plane-parallel radiative transfer, and tend to overestimate cloud effective radius, consequently underestimate N_d (Grosvenor et al., 2018). Therefore, the covariation of biases in N_d and AOD could partly offset the increase of N_d caused by increasing AOD, thus underestimate ACI. The previous study has illustrated that the covariation of retrieval biases in aerosol and cloud properties incurs a false correlation between them, thereby underestimating the cloud albedo effect (Jia et al., 2019b).

To obtain the collocated aerosol and cloud retrievals for statistical analysis, one has to collect

aerosol retrieval adjacent to clouds. Thus, the problem of biased AOD is rather difficult to be avoided for satellite-based investigations on aerosol-cloud interactions.

The relatively large response over land (compared to near-by oceans) is argued that the “anthropogenic AOD fraction is much larger over land” and nicely documented in Figure 4. I really appreciate this plot, because it is so important for the derived ACI impact, because a high anthrop. fraction determines not only the anthropogenic increase but also the pre-industrial background. The now applied anthropogenic AOD and AOD_f fractions, however, are very high in NH industrial regions, compared to what AeroCom modeling suggests (see figure below), especially if a 1850 reference year is applied.

Thanks for the comment. We agree that anthropogenic fraction plays a critical role on the RF_{aci} estimate. In the revised manuscript, we thus adopt anthropogenic fractions of AOD_f from the simulations of AeroCom phase1 and phase2 models used in MACv2 aerosol climatology products (Kinne et al., 2019), in addition to GEOS-Chem-APM modeled ones, to estimate RF_{aci} . The result shows that with the change of anthropogenic fractions of AOD_f, the global averages and spatial distributions of RF_{aci} change accordingly (Figure S9 and Table S3), implying the high sensitivity of RF_{aci} to the choice of anthropogenic fraction, which highlights the need for constraining anthropogenic fraction. The associated discussions have been inserted in section 2.3 (lines 277-288).

While anthropogenic fraction is quite revealing, it does not tell the entire story, as it is only a relative property, whereas the product (with AOD and AOD_f) matters. Thus, a map showing the pre-industrial AOD or AOD_f and the anthropogenic AOD or AOD_f would very insightful (if not in the main text, so at least in the supplement). In fact the anthropogenic fraction difference in Figure 4 and the is not so interesting.

Thanks for the suggestion. The spatial distributions of pre-industrial AOD (AOD_f) and the anthropogenic AOD (AOD_f) have been added to the supplement (Figure S6), and the map of anthropogenic fraction difference has been removed from the revised manuscript.

I agree that the impact from the solar insolation is relatively minor (although it will certainly matter for the winter season over Europe).

We agree with the reviewer.

I really like that the authors now also work with the AOD_f (which is a better proxy for aerosol number than AOD). By the way the authors also applied AI (should be similar in response to AOD_f, except that Angstrom values are usually terrible especially from retrievals) and sulfate mass (sulfate is the most likely droplet nuclei) in the Appendix and demonstrated differences to slopes using just AOD – and these differences should largely mimic AOD_f vs AOD based slope differences. With respect to comparing AOD_f and AOD, the much lower slope in Figure 2a over the northern hemispheric industrial regions by POLDER data makes me wonder, because for ‘cld’ with Merra in Figure 3c these are the regions with the largest ACI contributions.

The reason for the lower slopes over the northern hemispheric industrial regions by POLDER data compared to MERRA-2 data, is similar as that for much lower slopes for *AeroCld_Modis* case. As discussed in the main text, one reason is that satellite retrieval biases on both aerosol

and cloud tend to underestimate the slopes over lands (Jia et al., 2019b). More importantly, the analysis with POLDER data also suffers from the problem of sampling biases, which is more severe over the northern hemispheric industrial regions (Figure 1).

Also by replacing AOD with AOD_f in the analysis the estimated ACI impact for total AOD in Figure 3d (-0.67 global) almost doubles (-1.22 global) in Figure 4d, mainly, as I see it, with significant larger contributions over NH oceans, although there are now larger contributions over continents as well. Since ACI impacts increased everywhere, I wonder if this is possibly related to a cleaner pre-industrial background, by applying pre-industrial AOD_f rather than pre-industrial AOD?

Yes, applying the pre-industrial AOD_f rather than total AOD, would set a cleaner natural background as shown in Figure S6. Since the fine-mode aerosols dominate the anthropogenic contributions, with almost the same absolute increase of AOD and AOD_f from PI to PD, one obtains a much larger anthropogenic fraction if AOD_f rather than total AOD is applied, thereby a larger RF_{aci} , which has been discussed in the main text (lines 224-226).

With respect to differences in slopes between land and ocean, I think that aerosol number is the most important element and other properties (e.g. chem composition) at least for the fine-mode are only secondary. Interestingly, the regional slope comparisons in Figure 2 show a very strong inter-land and inter-ocean region variability, whereas on average all slopes over land and ocean are not so much different.

Thank for the comment. We agree that the differences in aerosol chemical components would not be that important if the fine-mode AOD rather than total AOD is applied. As shown in Figures 2c & d, the averaged slopes over lands (0.21 for POLDER-3 case and 0.35 for MERRA-2 case) and oceans (0.20 for POLDER-3 case and 0.34 for MERRA-2 case) are close when using AOD, which is consistent with what the reviewer pointed out. But when AOD_f is applied, the averaged slopes over lands (0.16 for POLDER-3 case and 0.37 for MERRA-2 case) become much lower than that over oceans (0.25 for POLDER-3 case and 0.42 for MERRA-2 case).

Thanks for comparing daily slopes with monthly slopes in Figure R2. I would interpret the results, that the monthly slopes are usually larger over the important northern hemispheric land and ocean regions for the Pacific and Asia (as I had expected) except for the more northern located Atlantic and Europe, where missing winter data may contribute (although during high latitude winters - when there is little to no sun - ACI contributions should be small).

Thanks for this explanation. Also, the monthly-based analysis suggested by the reviewer has been included in the revised manuscript (see the response above for more details).

The mentioned concerns on applying monthly statistics are noted. I still think though that such an analysis has some merit. Satellite retrievals now cover almost from 2 decades a potentially poor statistical aspect not be such a concern, especially if applied to larger regions. And while it is clear that more instantaneous data have more variability, local ACI impacts are often overestimates (e.g. shiptracks) and ACI responses are not always immediate often with delays of several days, so monthly averages possibly could better capture the 'effective' response.

Thanks for the suggestion. The monthly-based analysis has been included in the revised

manuscript (see the above response for more details), which is a key change in the revision.

Supplement:

The figure below in the left column presents anthropogenic AODf fractions from AeroCom modeling, generally larger fractions with an older (year 1750, AeroCom 1 emissions) reference and smaller fractions with a more recent (year 1850, AeroCom 2 emissions) reference. Using the 1850 reference (lower left) anthropogenic AODf contributions over the industrial areas are much lower than for the 1750 reference and much lower than those assumed by the authors in Figure 4b. A lower anthropogenic AODf fraction also means a much stronger pre-industrial background, and both elements reduce ACI impacts. In summary, without confidence in the anthropogenic AODf fraction large ACI uncertainties will remain.

MACv2: anthropogenic fraction of AODf today- AODf 1750 (AeroCom1 models, upper left) and anthropogenic fraction of AODf today- AODf 1850 (AeroCom2 models, lower left), anthropogenic BC fraction (upper right) and anthropogenic dust fraction (lower right)

We greatly appreciate this comment. We agree strongly with the reviewer that the uncertain anthropogenic definition drives the current uncertainty in the RF_{aci} estimate. Therefore, we have now in the revision also applied anthropogenic AOD_f fractions from AeroCom phase1 and phase2 modeling as proposed by the reviewer, in addition to the GEOS-Chem-APM result, to estimate the RF_{aci} . With the adequate CCN proxy, and meanwhile, sidestepping the sampling biases, the estimated RF_{aci} range from -1.02 to -1.68 $W m^{-2}$ when applying different anthropogenic fractions (Table S3). The result highlights the need for more meaningfully constrained anthropogenic fraction. It should be noted that no matter which anthropogenic fraction is applied, the impact of the sampling biases (the focus of this study), is obviously

substantial. The associated discussions have been inserted in section 2.3 (lines 277-288).

Minor comments

160 using Polder is a good idea, but there are limited years (MISR also provides AOD_f globally for the entire Terra data record ... but is spatially sparse). Still in the analysis simulated MERRA AOD_f were preferred, which clouds the entire process. Have there been MERRA data comparisons at successful Polder retrievals? Do they statistically agree?

Since the sampling biases can greatly affect the POLDER-3 daily-based analysis, the RF_{aci} was still estimated based on MERRA-2 AOD_f in the previous revision. However, in the current revision, we have also included the RF_{aci} estimate based on monthly POLDER data as suggested by the reviewer.

A comparison between POLDER-3 and MERRA-2 AOD_f demonstrates that they have a good agreement with each other (Figure S12), which has been mentioned in the revised manuscript (Lines 367-368).

163 ... why 30% of the seasalt AOD in the MERRA AOD_f? Is there a good reason?

AOD_f is not currently provided in MERRA-2 standard product. As suggested by Bellouin et al. (2013), the fine-mode fraction of sea-salt aerosols is also assumed to be 0.3 in our study, which is consistent with the size distributions assumed in MACC re-analysis product, where 30 % of globally-averaged sea-salt AOD is due to the two bins below 0.5 μm in the model. The study by Bellouin et al. (2013) has been cited in the manuscript (line 166).

References

- Bellouin, N., Quaas, J., Morcrette, J. J., & Boucher, O. (2013). Estimates of aerosol radiative forcing from the MACC re-analysis.
- Chen, J., Liu, Y., Zhang, M., & Peng, Y. (2016). New understanding and quantification of the regime dependence of aerosol-cloud interaction for studying aerosol indirect effects. *Geophysical Research Letters*, 43, 1780–1787. <https://doi.org/10.1002/2016GL067683>
- Gryspeerd, E., & Stier, P. (2012). Regime- based analysis of aerosol- cloud interactions. *Geophysical Research Letters*, 39, L21802. <https://doi.org/10.1029/2012GL053221>
- Grosvenor, D. P., Sourdeval, O., Zuidema, P., Ackerman, A., Alexandrov, M. D., Bennartz, R., et al. (2018). Remote sensing of droplet number concentration in warm clouds: A review of the current state of knowledge and perspectives. *Reviews of Geophysics*, 56, 409–453. <https://doi.org/10.1029/2017RG000593>
- Hudson, J. G., & Noble, S. (2014a). CCN and vertical velocity influence on droplet concentration and supersaturations in clean and polluted stratus clouds. *Journal of the*

Atmospheric Sciences, 106, 24,119–24,126.

Hudson, J. G., & Noble, S. (2014b). Low altitude summer/winter microphysics, dynamics and CCN spectra of northeastern Caribbean small cumuli; and comparisons with stratus. *Journal of Geophysical Research: Atmospheres*, 119. 5445–5463
<https://doi.org/10.1002/2013JD021442>

Jia, H., Ma, X., Yu, F., Liu, Y., & Yin, Y. (2019a). Distinct impacts of increased aerosols on cloud droplet number concentration of stratus/stratocumulus and cumulus. *Geophysical Research Letters*, 46. <https://doi.org/10.1029/2019GL085081>

Jia, H., Ma, X., Quaas, J., Yin, Y., & Qiu, T. (2019b). Is positive correlation between cloud droplet effective radius and aerosol optical depth over land due to retrieval artifacts or real physical processes?. *Atmospheric Chemistry & Physics*, 19 (13).

Kinne, S. (2019). The MACv2 aerosol climatology. *Tellus B: Chemical and Physical Meteorology*, 71(1), 1-21.

Várnai, T., & Marshak, A. (2009). MODIS observations of enhanced clear sky reflectance near clouds. *Geophysical Research Letters*, 36 (6).

Zhang, J., Reid, J., and Holben, B.: An analysis of potential cloud artifacts in MODIS over ocean aerosol optical thickness products, *Geophys. Res. Lett.*, 32, L15803, <https://doi.org/10.1029/2005GL023254>, 2005.

Reviewer #2 (Remarks to the Author):

I am satisfied that the authors have addressed all of my previous concerns and am happy to accept the paper in its current form.

We thank the reviewer for taking the time to assess the manuscript.

Reviewer #3 (Remarks to the Author):

I thank the authors for their careful consideration of my comments and clarifications to their document. I am satisfied with their responses.

There seems to be a typesetting issue on page 5 with the entire page in heading font.

Thanks for the reminder. This typesetting issue has been fixed in the revised manuscript.

Review of re-revised paper:

Significant underestimation of radiative forcing by aerosol-cloud interactions derived from satellite based methods

by H. Jia et al.

General comments

I really appreciate that the authors addressed my concerns – and then in this detail. Thanks.

However, to start my response, I am still struggling with the explanation by the authors that in observational associations missed cases due to cloud cover (and ‘possibly’ higher COT cases) should exert a stronger (missed) cooling, because (as they argue) “optically thicker clouds exert in an absolute sense a stronger cooling than optically thinner clouds”. For ACI, we are examining changes (!) so that the (first indirect effect) susceptibility (to the planetary albedo) contradicts this line of reasoning. For me - a better argument would be a likely more humid environment associated with the missed higher cloud cover cases. A higher relative humidity would allow cases for extra ACI cooling by an extended cloud lifetime (and with delayed precip Nd stays longer high) whereas at a lower ambient relative humidity, in cases where aerosol is easier observable, smaller droplets can evaporate faster by dry air entrainment (Nd decreases).

The extra analysis with monthly associations is a nice addition. I understand that the grey columns in Figure R1 are based on observational associations (no Merra2 data correct? if so are Nd from MODIS?). These regional (grey column) slopes indeed are consistently lower than the regional ‘complete’ Merra2 based slopes (dark blue columns). But the differences are small (~ - 10%) – especially over (dark background) oceans, where ACI impacts dominate. Also again, the interpretation will be limited to regions without significant mineral dust (and seasalt) contributions as AOD and not AODf is here applied as reference. I also do not quite understand the AOD clear-sky scenario. Clear-sky cases ($f=0$) over entire 1x1deg regions are relatively rare. For clear-sky AOD I also would have considered AOD from clear-sky sub-regions (AOD retrievals are usually offer AOD at 10x10km²)...?

The Figure R2 uses AODf as a reference (great) - but it raises more questions. In order to demonstrate that monthly relationships agree or disagree with daily relationships it would perhaps have been better to use for monthly and daily data the same data source (e.g. only POLDER or only Merra2 ... and I assume that the applied Nd data are from Merra2 ...?). But also here the slopes for the ocean regions are quite similar. However, the regional slope differences over land are at times very large and a bit puzzling to me.

Now comparing the slopes of Figure R1 (monthly associations) and Figure 2 (daily associations) is interesting and should deserve some comments. Correct me, if I am wrong, but I conclude that the slope differences between ‘retrieval’ observations and Merra (all cases) modeling are smaller for monthly data than for daily data and that the regional slopes are usually slightly larger for monthly than for daily associations. Anyway, this seems an interesting result and deserves some mentioning and discussion in the paper.

With respect to uncertain small values for AOD and AODf values (especially with respect to $\ln Nd/\ln AOD$ relationships) it is nice to see that the lowest AOD/ AODf cases have been removed – also as at almost identical values (within errors) no useful associations should be drawn. When applying an AOD lower threshold (especially involving satellite retrieved AOD) I would not work with a (15) % values but rather with an absolute value (e.g. no AOD_{550nm} below 0.04).

In addition, I appreciate the detailed responses and explanation to my concerns and comments.

Maybe at the end I should share some initial association results from on-going studies with retrieval data (monthly MODIS AOD_f and AVHRR CDNC averages for 16 years). The associations confirm that slopes for $\ln(\text{AOD}_f)/\ln(\text{CDNC})$ are indeed significantly larger in more cloudy environment (I compared 1x1 regions - where on a statistically basis - low altitude cloud cover exceeded 0.6 vs stayed below 0.3 over oceans and cloud cover exceeded 0.4 vs stayed 0.2 over land). If only considering AOD_f values between 0.06 and 0.21 for monthly associations to CDNC in the same 1x1 deg regions then I get larger slopes over land (.438) than over oceans (.277) in the regions with pre-existing significant cloud cover. But over 1x1 regions, where few clouds are expected, the slopes are much smaller not only over oceans (0.095) but especially over land (0.008). I am still struggling with the explanation of these weak slopes (semi-direct effects?) if fewer clouds are involved but in tendency these results indirectly confirm the author's claim that the missed pixels (due to cloud cover) in daily associations likely contribute to ACI underestimates.

In summary, the paper with so much interesting data-analysis and interpretation efforts deserves a publication and I hope the authors continue their investigations.

Thanks to the reviewer for providing thoughtful comments and suggestions to improve the manuscript. Below we address the reviewer's comments, with the reviewer comments in black, and our responses in blue. We have revised the manuscript accordingly.

Review of re-revised paper:

Significant underestimation of radiative forcing by aerosol-cloud interactions derived from satellite based methods

by H. Jia et al.

General comments

I really appreciate that the authors addressed my concerns – and then in this detail. Thanks. However, to start my response, I am still struggling with the explanation by the authors that in observational associations missed cases due to cloud cover (and 'possibly' higher COT cases) should exert a stronger (missed) cooling, because (as they argue) "optically thicker clouds exert in an absolute sense a stronger cooling than optically thinner clouds". For ACI, we are examining changes (!) so that the (first indirect effect) susceptibility (to the planetary albedo) contradicts this line of reasoning. For me - a better argument would be a likely more humid environment associated with the missed higher cloud cover cases. A higher relative humidity would allow cases for extra ACI cooling by an extended cloud lifetime (and with delayed precip Nd stays longer high) whereas at a lower ambient relative humidity, in cases where aerosol is easier observable, smaller droplets can evaporate faster by dry air entrainment (Nd decreases).

Thanks for the comment. We fully agree with the reviewer that the stronger cooling should not be linked to higher COT (i.e., optically thicker clouds) since ACI is related to the relative change rather than the absolute change. As replied in the previous response, the COT-associated results have been already removed, and "optically thicker clouds exert in an absolute sense a stronger cooling than optically thinner clouds" as mentioned by the reviewer is no longer our argument. The increased slope by including all missed clouds is likely due to the presence of more stratiform clouds, which have been reported to be more sensitive to aerosol perturbations than cumuliform clouds (Gryspeerd & Stier., 2012; Vogelmann et al., 2012; Jia et al., 2019a). Stratiform clouds are typically characterized by high cloud fraction, thus a larger slope would occur with a higher cloud fraction. We thank the reviewer for the interpretation regarding the relative humidity. This is very interesting and also consistent with our argument of cloud type to an extent since stratiform clouds tend to exist in a moister environment than cumuliform ones.

The extra analysis with monthly associations is a nice addition. I understand that the grey columns in Figure R1 are based on observational associations (no Merra2 data correct? if so are Nd from MODIS?). These regional (grey column) slopes indeed are consistently lower than the regional 'complete' Merra2 based slopes (dark blue columns). But the differences are small (~ - 10%) – especially over (dark background) oceans, where ACI impacts dominate. Also again, the interpretation will be limited to regions without significant mineral dust (and seasalt) contributions as AOD and not AODf is here applied as reference. I also do not quite

understand the AOD clearsky scenario. Clear-sky cases ($f=0$) over entire 1x1deg regions are relatively rare. For clear-sky AOD I also would have considered AOD from clear-sky sub-regions (AOD retrievals are usually offer AOD at 10x10km²)...?

Thanks for the comment. All results in Figure R1 are based on MODIS-Nd and MERRA2-AOD associations, but with four different subsampling strategies as indicated by the subscripts of *AllCld*, *AeroCld*, and *clear sky*. Actually, the relative increases of slopes from *AeroCld* (grey columns) case to *AllCld* ('complete' Merra2 based slopes as mentioned by the reviewer) case are generally larger than 20 % (20 % for NPO, 38 % for NAO, 34 % for SPO, 23 % for SAO, and 26 % for SIO) except for three tropical oceanic regions (4 % for TPO, 19 % for TAO, and -1.5 % for TIO), which confirms that the sampling biases can significantly influence ACI as also demonstrated by our daily-based results.

In order to better interpret the results over regions with significant mineral dust and/or seasalt aerosols, we also conduct the same analysis as in Figure R1 but adopting MERRA2 AODf (Figure S8 in the revised manuscript). The conclusion is consistent with that based on total AOD.

Here the 'clear-sky' case includes not only the AOD with $f = 0$ but also the AOD without successful N_d retrieval over entire 1x1deg regions, which is used to represent the monthly AOD that is irrelevant to monthly $N_{dAllCld}$. The definition of 'clear-sky' case has been clarified in the caption of Figure S8 in the revised manuscript.

The Figure R2 uses AODf as a reference (great) - but it raises more questions. In order to demonstrate that monthly relationships agree or disagree with daily relationships it would perhaps have been better to use for monthly and daily data the same data source (e.g. only POLDER or only Merra2 ... and I assume that the applied Nd data are from Merra2 ...?). But also here the slopes for the ocean regions are quite similar. However, the regional slope differences over land are at times very large and a bit puzzling to me.

As replied in the previous response, the comparison in Figure R2 is not to demonstrate whether monthly relationships agree or disagree with daily relationships, but to explain the much smaller R_{FacI} for the POLDER-3-monthly case than for the MERRA-2-daily case over land (Figure 4c & d), which are considered as two potential ways to overcome the sampling biases and obtain more accurate R_{FacI} estimates. All cloud property (including N_d) data used in this study are from MODIS retrievals (see *Methods*). MERRA-2 re-analysis data are used just to provide aerosol information.

With respect to the much lower slopes for the POLDER-3 monthly case than the MERRA-2 daily case over most land regions but similar slopes over ocean regions, one reason is that monthly AODf (still missing aerosol information when clouds fully cover 1x1deg regions) would not be well representative of the amount of aerosols actually linking with monthly N_d (including all clouds), especially over the land regions where aerosols have large inter-daily variability. Another reason would be the satellite retrieval uncertainty in the POLDER-3 case, which has been found to significantly underestimate the ACI especially over land (Gryspeerd & Stier, 2012; Jia et al., 2019b). The associated discussions have already been included (lines 226-234).

Now comparing the slopes of Figure R1 (monthly associations) and Figure 2 (daily associations)

is interesting and should deserve some comments. Correct me, if I am wrong, but I conclude that the slope differences between 'retrieval' observations and Merra (all cases) modeling are smaller for monthly data than for daily data and that the regional slopes are usually slightly larger for monthly than for daily associations. Anyway, this seems an interesting result and deserves some mentioning and discussion in the paper.

Thanks for the suggestion. The result in Figure R1 is based on MERRA-2 AOD, so it does not include any pure 'retrieval' observations. With respect to the larger slopes for monthly than for daily analyses, the associated discussion has been included in the revised manuscript (lines 234-236) as suggested by the reviewer.

With respect to uncertain small values for AOD and AODf values (especially with respect to $\ln AOD/\ln AODf$ relationships) it is nice to see that the lowest AOD/ AODf cases have been removed – also as at almost identical values (within errors) no useful associations should be drawn. When applying an AOD lower threshold (especially involving satellite retrieved AOD) I would not work with a (15) % values but rather with an absolute value (e.g. no AOD,550nm below 0.04).

Thanks for the comment. In this study, it is difficult to accurately determine the thresholds of both AOD and AODf for MERRA-2 re-analysis and different satellite retrievals (MODIS and POLDER-3), representatively, which would introduce additional uncertainties when comparing these results. Therefore, to meaningfully compare and interpret the results based on different CCN proxies from different data sources, a relative threshold (the lowest 15%) proposed by Hasekamp et al. (2019) was used here. But for the analysis only involving a single CCN proxy from the same data source, we entirely agree with the reviewer that applying an absolute threshold value would be better to constrain the uncertainty for low AOD retrieval.

In addition, I appreciate the detailed responses and explanation to my concerns and comments. Maybe at the end I should share some initial association results from on-going studies with retrieval data (monthly MODIS AODf and AVHRR CDNC averages for 16 years). The associations confirm that slopes for $\ln(AODf)/\ln(CDNC)$ are indeed significantly larger in more cloudy environment (I compared 1x1 regions - where on a statistically basis - low altitude cloud cover exceeded 0.6 vs stayed below 0.3 over oceans and cloud cover exceeded 0.4 vs stayed 0.2 over land). If only considering AODf values between 0.06 and 0.21 for monthly associations to CDNC in the same 1x1 deg regions then I get larger slopes over land (.438) than over oceans (.277) in the regions with pre-existing significant cloud cover. But over 1x1 regions, where few clouds are expected, the slopes are much smaller not only over oceans (0.095) but especially over land (0.008). I am still struggling with the explanation of these weak slopes (semi-direct effects?) if fewer clouds are involved but in tendency these results indirectly confirm the author's claim that the missed pixels (due to cloud cover) in daily associations likely contribute to ACI underestimates.

We really appreciate these interesting results. With respect to the weaker slopes under low cloud cover condition, another possible explanation would be the large retrieval bias for broken clouds (often associated with low cloud cover). Our previous study (Jia et al., 2019b)

found that the covariation of retrieval biases in Nd (due to partly cloudy condition) and AOD (cloud contaminations and cloud adjacency effect) can greatly underestimate ACI. Thus if pure satellite observations are used to derive ACI, which suffers from the covariation of retrieval biases, the problem of retrieval biases might deserve some attention.

In summary, the paper with so much interesting data-analysis and interpretation efforts deserves a publication and I hope the authors continue their investigations.

Thanks so much to the reviewer for taking the time to assess the manuscript and for providing a lot of thoughtful and helpful comments and suggestions, which have allowed us to clarify and improve the manuscript. The reviewer's comments also inspire us to do further exploration on this topic. Many thanks!

Reference

Gryspeerd, E. & Stier, P. Regime-based analysis of aerosol-cloud interactions. *Geophys. Res. Lett.* 39, L21802 (2012).

Hasekamp, O. P., Gryspeerd, E., & Quaas, J. Analysis of polarimetric satellite measurements suggests stronger cooling due to aerosol-cloud interactions. *Nat. Commun.* 10(1), 5405 (2019).

Jia, H., Ma, X., Yu, F., Liu, Y. & Yin, Y. Distinct impacts of increased aerosols on cloud droplet number concentration of stratus/stratocumulus and cumulus. *Geophys. Res. Lett.* 46, 13517–13525 (2019a).

Jia, H., Ma, X., Quaas, J., Yin, Y. & Qiu, T. Is positive correlation between cloud droplet effective radius and aerosol optical depth over land due to retrieval artifacts or real physical processes? *Atmos. Chem. Phys.* 19, 8879–8896 (2019b).

Vogelmann, A. M., McFarquhar, G. M., Ogren, J. A., Turner, D. D., Comstock, J. M., Feingold, G., et al. RACORO extended-term aircraft observations of boundary layer clouds. *Bull. Am. Meteorol. Soc.*, 93(6), 861–878 (2012).

Review of the re-re-revised paper:

Significant underestimation of radiative forcing by aerosol-cloud interactions derived from satellite based methods

by H. Jia et al.

General comments

Thanks for the detailed answers to my comments.

I am particular happy to hear that the 'optically thicker cloud' argument has been removed.

And I really like the monthly association analysis, although it is in the appendix (i.e. Figure S8). With careful reading and your extra explanations I also now understand the 'clear-sky' AOD (AODf) monthly associations, which include all available daily aerosol data within a 1x1 grid (even if there are NO matching MODIS Nd cloud data available on a particular day). Thus, this 'allcld/clear-sky' case (lightest blue in Figure S8) mimics monthly associations of successful AOD(AODf) retrievals and successful MODIS Nd retrievals. In contrast, the 'aero-cld/aero-cld' case (grey color in Figure S8) only considers in the monthly relationship cases with local daily associations. Thus, by comparing the two cases, Figure S8 actually demonstrates that Nd responses to AODf for both data-selections are similar in strength for most regions (while non-agreeing lower monthly responses for SAO and SAM can probably be linked to lofted non-interacting wildfire aerosol). Figure S8 also shows the impact by adding more associations with the added inclusion of MERRA AOD(AODf) data in cloudy regions (where no aerosol retrievals are offered – mainly due to overcast conditions), This results in a stronger Nd response (consistent with the theme of the paper), but also to part in a general trust in MERRA interpretations / distributions of AOD (AODf) in cloudy regions, where cloud processing (e.g. removal and/or swelling) will introduce uncertainties. Hereby Figure S8 shows that the Nd response increases are weaker (only on the order of 10%) for (the more relativistic) AODf than for AOD (for which the percentage increases were calculated and stated by the authors).

With respect to the POLDER data, I agree that there many open AOD retrieval issues especially over continents, although only relative changes are investigated here.

For the other comments / responses there was no really a disagreement. It is now time to publish the paper and let the authors continue and intensify their studies on aerosol-cloud interaction observational constraints.

By the way, the case definitions (even listed Table1) still seems a bit confusing as they are applied to both Nd and AOD(AODf) data. A visual example (I am more a visual person) would go a long way (e.g. different color point/pixels within a few 1x1 fields for aerosol and clouds) to convey with different color combinations the different cases.

Thanks to the reviewer for providing thoughtful comments and suggestions to improve the manuscript. Below we address the reviewer's comments, with the reviewer comments in black, and our responses in blue. We have revised the manuscript accordingly.

Review of the re-re-revised paper:

Significant underestimation of radiative forcing by aerosol-cloud interactions derived from satellite based methods

by H. Jia et al.

General comments

Thanks for the detailed answers to my comments.

I am particular happy to hear that the 'optically thicker cloud' argument has been removed.

And I really like the monthly association analysis, although it is in the appendix (i.e. Figure S8). With careful reading and your extra explanations I also now understand the 'clear-sky' AOD (AOD_f) monthly associations, which include all available daily aerosol data within a 1x1 grid (even if there are NO matching MODIS Nd cloud data available on a particular day).

Thus, this 'allcld/clear-sky' case (lightest blue in Figure S8) mimics monthly associations of successful AOD(AOD_f) retrievals and successful MODIS Nd retrievals. In contrast, the 'aero-cld/aero-cld' case (grey color in Figure S8) only considers in the monthly relationship cases with local daily associations. Thus, by comparing the two cases, Figure S8 actually demonstrates that Nd responses to AOD_f for both data-selections are similar in strength for most regions (while nonagreeing lower monthly responses for SAO and SAM can probably be linked to lofted noninteracting wildfire aerosol). Figure S8 also shows the impact by adding more associations with the added inclusion of MERRA AOD (AOD_f) data in cloudy regions (where no aerosol retrievals are offered – mainly due to overcast conditions), This results in a stronger Nd response (consistent with the theme of the paper), but also to part in a general trust in MERRA interpretations / distributions of AOD (AOD_f) in cloudy regions, where cloud processing (e.g. removal and/or swelling) will introduce uncertainties. Hereby Figure S8 shows that the Nd response increases are weaker (only on the order of 10%) for (the more relativistic) AOD_f than for AOD (for which the percentage increases were calculated and stated by the authors).

Thanks for the discussion. We do agree with the reviewer. With respect to the use of MERRA-2, we have validated the re-analyzed AOD with the satellite data used in this study (e.g., MODIS and POLDER-3 AOD/AOD_i), and demonstrated that they generally agree well with each other. The systematically larger satellite-retrieved AOD than re-analyzed AOD in partly cloudy regions would be largely attributed to retrieval problems rather than the MERRA-2 side. We also agree that cloud processing will introduce uncertainties, which has been already mentioned in discussion section.

With respect to the POLDER data, I agree that there many open AOD retrieval issues especially over continents, although only relative changes are investigated here.

We agree with the reviewer.

For the other comments / responses there was no really a disagreement. It is now time to publish the paper and let the authors continue and intensify their studies on aerosol-cloud interaction observational constraints.

Thanks so much to the reviewer for providing helpful comments and suggestions.

By the way, the case definitions (even listed Table1) still seems a bit confusing as they are applied to both Nd and AOD(AODf) data. A visual example (I am more a visual person) would go a long way (e.g. different color point/pixels within a few 1x1 fields for aerosol and clouds) to convey with different color combinations the different cases.

Thanks for the suggestion. We have now replaced Table 1 with a visual schematic plot (Figure 1) in the revised manuscript.